# Team, Then Trim: An Assembly-Line LLM Framework for High-Quality Tabular Data Generation

## Abstract

While tabular data is fundamental to many real-world machine learning (ML) applications, acquiring high-quality tabular data is usually labor-intensive and expensive. Limited by the scarcity of observations, tabular datasets often exhibit critical deficiencies, such as class imbalance, selection bias and low fidelity. To address these challenges, building on recent advances in Large Language Models (LLMs), this paper introduces team-then-trim, a framework that synthesizes high-quality tabular data through a collaborative team of LLMs, followed by a rigorous data quality control (QC) pipeline. In our framework, tabular data generation is conceptualized as a manufacturing process: specialized LLMs, guided by domain knowledge, are tasked with generating different data components sequentially, and the resulting products, i.e., the synthetic data, are systematically evaluated across multiple dimensions of QC. Empirical results on both simulated and real-world datasets demonstrate that our framework outperforms the state-of-the-art methods in producing high-quality tabular data, highlighting its potential to support downstream models when direct data collection is practically infeasible.

## 1 Introduction

Machine learning (ML) systems in domains like healthcare (Provost & Murray, 2022), transportation (Washington et al., 2020), and the social sciences (Aneshensel, 2012) depend heavily on tabular datasets. According to uniform convergence theorems and generalization bounds in classical statistical learning theory (Feldman & Vondrak, 2018), if we can sample a sufficiently large number of tabular data points from the underlying space, we can reliably train predictive models that generalize well and achieve accurate performance. In practice, however, we often have to begin with a tabular dataset sampled from the true data distribution, which is typically small and may suffer from various deficiencies, such as bias (Little & Rubin, 2019), imbalance (Thabtah et al., 2020) and noise (Gupta & Gupta, 2019). For example, in medical studies of diseases such as type-I diabetes, patients may constitute only a small minority compared to non-patient participants, while all of them are at some level of risk as indicated by the screening results. Under such data imbalance, downstream models trained solely on a limited original dataset often struggle to capture the true decision boundary, leading to biased predictions, poor generalization to minority groups, and reduced robustness(Chen et al., 2023). If we can generate high-quality data that are plausible under domain constraints and diverse enough to cover rare or unseen configurations, the accuracy and robustness of the downstream models can be improved without costly data collection.

However, traditional data generation methods, such as resampling (e.g., SMOTE (Chawla et al., 2002)) can only extrapolate from observed samples. While effective within the boundaries of the original data, they remain constrained in exploration, often reinforcing the same biases and failing to recover rare or missing subpopulations (Li & Vasconcelos, 2019). Deep generative models, e.g., CTGAN, TVAE (Xu et al., 2019), although more expressive, typically demand substantial amounts of data and struggle in low-data regimes where augmentation is most needed. More recently, Large Language Models (LLMs) have emerged as a new paradigm for data generation, leveraging broad world knowledge and few-shot reasoning abilities (Seedat et al., 2023; Patel et al., 2024). However, a single LLM used in isolation is prone to inconsistencies: it ignores structural dependencies among features, violates logical constraints, and cannot holistically capture the full complexity of

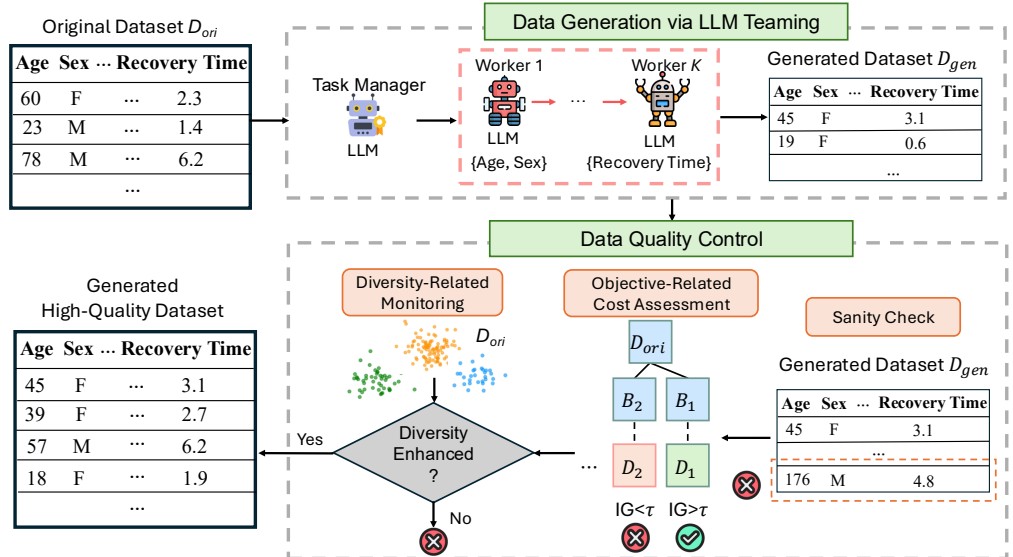

Figure 1: An illustration of our team-then-trim framework. The LLM teaming is for structured data generation and the trimming is for rigorous three-stage QC.

tabular datasets. Further related work is provided in Appendix A. These issues limit their reliability, especially when tasked with generating multi-faceted records from small seed datasets.

Beyond generation, ensuring the quality of synthetic data is equally critical. LLMs are known to hallucinate (Bang et al., 2025), introduce implausible entries (Yao et al., 2023), or produce distributions that deviate from domain requirements (Xu et al., 2024). Existing methods typically focus on plausibility checks or heuristic filtering (Sousa et al., 2024), but they often overlook deeper dimensions of quality, such as consistency with downstream objectives, fidelity to minority subgroups, and diversity of coverage. Without a principled quality control (QC) process, synthetic data risks amplifying noise rather than mitigating it.

To address these two challenges, in this paper, we introduce *team-then-trim*, a framework that combines teaming for structured data generation with trimming for rigorous three-stage QC. Unlike monolithic generators, our framework decomposes a dataset into semantically coherent components and assigns each to a specialized LLM worker with a clearly defined role. Workers operate sequentially, conditioning on previously produced components to preserve inter-feature logic and domain constraints. Team-then-trim produces data in batches, allowing large volumes of synthetic data to be created with lower cost and higher efficiency (Citovsky et al., 2021; Ren et al., 2021), while subjecting each batch to a rigorous QC process. The QC pipeline consists of three stages: (1) a sanity check validates variable ranges, categorical consistency, and inter-feature dependencies; (2) objective-related cost assessment, a model-based duel between LLM-generated batches and bootstrap samples from the original data, keeping low-residual points, and (3) diversity-related monitoring with clustering-based coverage checks that admits batches expanding coverage without skew. Together, these stages transform an initial synthetic pool into a task-aligned, diverse, and constraint-consistent dataset for downstream learning. The QC pipeline is modular and can be plugged into any generative approach. To sum up, the contributions of this paper are three-fold:

- We introduce team-then-trim, a novel framework that conceptualizes data augmentation as a product manufacturing process, where raw augmented data is initially synthesized in batches through the collaborative assembly of multiple specialized LLM generators (teaming) and subsequently refined through a rigorous QC process (trimming), ultimately yielding high-quality tabular data.

- We design a three-stage data QC pipeline to transform the raw synthetic data batches into high-quality data with only a small set of true data. The pipeline systematically refines data quality along multiple dimensions including validity, learnability, informativeness, and diversity.

- We conducted extensive experiments on various simulated and real-world datasets to demonstrate that our framework can reliably generate tabular data of high quality across diverse evaluation metrics. Its consistent superiority over state-of-the-art data augmentation methods provides strong evidence for the utility of LLMs, when coupled with effective QC, in addressing challenging ML scenarios characterized by data scarcity, imbalance, and noise, etc.

## 2 TEAM-THEN-TRIM FRAMEWORK

We now describe our team-then-trim framework for generating high-quality tabular data that can be used by a broad spectrum of downstream predictive models. Consider the tabular data space $\mathcal{X} \times \mathcal{Y}$ equipped with a probability measure induced by the true data distribution $p_{true}$. Here, $\mathcal{X} = \mathcal{X}_1 \times \cdots \times \mathcal{X}_d$ denotes the $d$-dimensional feature space where $\mathcal{X}_j$ is the $j$-th feature dimension, and $\mathcal{Y}$ denotes the label space. Our framework starts with a small tabular dataset sampled from $p_{true}$, denoted by $D_{ori} = \{(\boldsymbol{x}_i, y_i)\}_{i=1}^n$, where $\boldsymbol{x}_i = (x_{i,1}, \cdots, x_{i,d}) \in \mathcal{X}$ and $y_i \in \mathcal{Y}$. It aims to generate an additional set of new data $D_{gen}$ without explicit knowledge of $p_{true}$ and combine it with the original data $D_{ori}$ to build a new dataset $D_{new} = D_{ori} \bigcup D_{gen}$, such that a downstream predictive model $y = f(\boldsymbol{x})$ trained on this augmented dataset $D_{new}$ attains improved generalization performance compared to training exclusively on $D_{ori}$. An overview of the team-then-trim framework is shown in Fig. 1. We will present details in the following subsections.

### 2.1 RAW DATA GENERATION VIA LLM TEAMING

An assembly line in manufacturing typically breaks down a complex production process into a sequence of smaller subtasks which are carried out by specialized workers under the coordination of a task manager. Analogously, the team-then-trim framework employs a team of pretrained LLMs, where a task manager LLM and multiple worker LLMs collaborate to produce a raw tabular dataset. Given a data generation task, the task manager LLM, guided by its domain knowledge of the specific task and analysis of the true data $D_{ori}$, first decomposes the data structure into multiple components. It then assigns the subtasks of generating these components to different worker LLMs and organizes their execution according to a specified topology. The worker LLMs perform their assigned tasks to generate the components separately, following the instruction given by the task manager LLM in an assembly-line fashion.

**Task manager.** Given the original dataset $D_{ori}$, the task manger LLM $\Phi$ is additionally prompted with the feature dictionary to facilitate the understanding of the task (see Appendix Fig. 6). Based on its understanding and domain knowledge, it will partition the full feature space $\mathcal{X}$ into $K$ disjoint components, i.e., $\mathcal{X} = \mathcal{X}_{I_1} \times \cdots \mathcal{X}_{I_K}$, where $\mathcal{X}_I = \prod_{i \in I} \mathcal{X}_i$ is the cartesian product of all the single feature spaces whose indices fall in $I$. Thus, the partition should satisfy $\bigcup_{k=1}^K I_k = \{1, \ldots, d\}$ and for $\forall k, l = 1, \ldots, K, k \neq l, I_k \cap I_l = \emptyset$. This partition divides the data generation workload into $K$ components, allowing different LLM workers to handle distinct parts of the task.

Here, each component represents a coherent semantic category, with all the features share some underlying semantic characteristics though not necessarily correlated. For example, suppose the objective is to predict the post-surgery recovery time (outcome) with four predictors: age, sex, hospital resources and sleep quality. As Fig. 2 shows, these predictors can be grouped into three categories, "age" and "sex" as demographic features, "hospital resources" as an independent environmental feature, and "sleep quality" as a feature potentially influenced by both demographic and environmental factors. Without incorporating such structural knowledge, the generated data

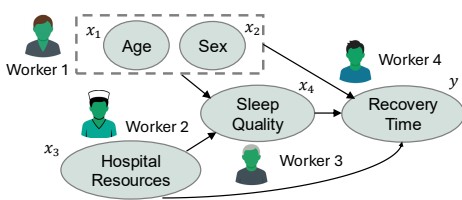

Figure 2: An example of task coordination.

may overlook the complex dependencies among features and fail to capture their semantically meaningful relationships. In our team-then-trim framework, the task manager LLM fully utilizes such knowledge to coordinate the generation task. It views the different data components as nodes on a graph, which results in an activity-on-vertex network (precedence graph (Boffey, 1982)) $G = (V, E)$ explicitly encoding their relationships where $V = \{I_1, \cdots, I_K\}$ and $E$ encodes the information of

the graph links. Thus, the work of the task manager LLM can be expressed as:

$$G = \Phi(\mathcal{P}(D_{ori}, \boldsymbol{c})), \tag{1}$$

where $\boldsymbol{c}$ is necessary contextual information of the dataset (e.g., variable dictionary) and $\mathcal{P}$ denotes the encoding function that turns the input information into prompts. Through this scheduling, components without inter-dependencies can be generated in parallel, whereas those with dependency constraints should be generated sequentially. With less information required for generating each component, the overall process is enabled to produce more precise and semantically coherent data.

**Specialized workers.** Our team-then-trim framework employs a set of LLMs $\Psi_1, \cdots, \Psi_K$ that act as specialized workers to efficiently and consistently generate the $K$ different data components defined by the task manager LLM. All the specialized workers follow the task coordination instructions provided by the task manager to generate data sequentially in the prescribed order. Each worker receives a prompt containing specific information about the component it is responsible for, along with all previously generated data components on which its assigned component may depend. The generated data component can be expressed as follows:

$$\mathbf{X}_k \sim \Psi_k(\mathcal{P}_k(D_{ori}, \boldsymbol{c}_k, G)| \bigcup_{l \in \text{pa}(I_k, G)} \mathbf{X}_l), \quad \forall k = 1, \cdots, K, \tag{2}$$

where $\mathcal{P}_k$ is the individual prompt encoding function, $\text{pa}(I_k, G)$ denotes the set of parent nodes of $I_k$ on graph $G$ and $\boldsymbol{c}_k$ is the contextual information about the features in the $k$-th component. Thus, the parent components, the generation schedule determined by the task manager and original data with contextual information together constitute the foundation for generating $\mathbf{X}_k$. For example, in Fig. 2, to generate the data column for the feature "sleep quality", worker 1 and 2 should generate the demographic feature component (including "age" and "sex") and the environmental feature component (including "hospital resources") beforehand, as it guarantees that the subsequent generation of "sleep quality" does not contradict the domain knowledge encoded within the LLM worker. It is important to note that, due to the stochastic nature of LLMs, the generation process is inherently not deterministic. Consequently, the generated data is not fixed even with identical prompts or conditions. Once the parent components have been generated by upstream LLM workers under the coordination of the task manager, the specialized worker proceeds to sample candidate data points from an underlying, unknown distribution as instructed by the received prompt. Since the features in the same component semantically belong to the same category, each worker can focus on generating semantically related concepts, which makes the generation process more efficient and reliable. Taking advantage of the prior knowledge, contextual understanding, and in-context learning capabilities of LLMs, the generated data is expected to exhibit higher internal consistency and factual fidelity (Saglam et al., 2025; Kozlowski et al., 2025; Wang et al., 2025) through the teaming of these workers, which is a notable advantage over a single LLM that handles multiple semantic categories simultaneously. An example that illustrates such advantage is shown in Appendix E. Finally, when all these workers finish their generation, we aggregate all the generated components and assemble them into a full design (unlabeled dataset) through a concatenation function:

$$\mathbf{X}_{gen} = \text{Concat}(\mathbf{X}_1, \cdots, \mathbf{X}_K). \tag{3}$$

An additional LLM worker $\Psi_{K+1}$ takes in the unlabeled dataset and generates labels for the design:

$$\boldsymbol{y}_{gen} \sim \Psi_{K+1}(\mathcal{P}_y(D_{ori}, \boldsymbol{c}, \mathbf{X})), \tag{4}$$

where $\mathcal{P}_y$ is the prompt encoding function. Here, the labels can be restricted to meaningful values by encoding the constraints in the contextual information $\boldsymbol{c}$. For example, for binary labeled data, the label worker is prompted to only produce values from $\{0, 1\}$. But if not required, the worker is allowed to generate different labels beyond the ones in the given original dataset to encourage its exploration. With the labels generated, combining all together, we can end up with a complete dataset $\mathcal{D}_{gen} = \{\mathbf{X}_{gen}, \boldsymbol{y}_{gen}\}$.

## 2.2 Data Quality Control

Given the original dataset $D_{ori}$ with necessary contextual information, a collaborative team of LLM workers coordinated by an LLM task manager can produce a dataset in an assembly-line fashion,

similar to manufacturing production. However, even though the generated data may appear plausible, it can still suffer from quality issues due to LLM hallucination or mistakes (Huang et al., 2025; Chen et al., 2024b; Liu et al., 2024). Just as physical products require QC to ensure their reliability, the raw products of the data generation assembly line, namely, the generated data, must also undergo a quality assurance process to ensure they are readily usable by various downstream applications. Essentially, it can be extremely hard to manipulate the generation process of the LLM workers given their black-box nature. Unless we perform full training or fine-tuning with a large amount of ground-truth data to guide the LLMs, both of which typically incur substantial cost, it is difficult to exert direct control over the quality of the generated data during the generation phase. Consequently, quality assurance efforts are preferably concentrated in the post-generation stage, which motivates the design of our three-stage plug-in QC pipeline.

In our pipeline, to further refine the raw data generated by the LLM assembly line in Section 2.1, QC is performed in a batch-wise manner, trimming or discarding data as necessary. Compared to checking the quality of a single data point, the use of the batch setting has multiple advantages, such as reducing cost, improving efficiency, etc. (Citovsky et al., 2021; Ren et al., 2021). Moreover, batches inherently provide richer collective information than isolated samples, which enables more effective and informative quality inspection at the batch level (Kirsch et al., 2019). Assume the batch size is $n_b$. The core idea of our QC process is: each time $t$ a small fixed-size dataset $D_t$ is generated from the LLM assembly line as a batch, its quality is rigorously evaluated. If the batch meets the predefined quality requirements, it is polished by trimming the unsatisfied samples in the batch and integrated into the existing dataset $D$ (initially $D_{ori}$) to construct a new set; otherwise, the full batch is discarded. This iterative generation-and-evaluation cycle continues until sufficient data has been admitted. For the evaluation, we ensure the batch quality from three complementary aspects: 1) Sanity check. The batch is examined on its variables to filter out apparently invalid or impossible data samples. 2) Objective-related cost assessment. Low-quality data is identified and rejected based on the batch's total cost related to the learning objective of a specific model. 3) Diversity inspection. The batch is further checked to ensure sufficient coverage in the full data space.

**Sanity check.** To ensure the samples in the batch $D_t$ is valid, we perform sanity check based on the types and values of features together with necessary relation constraints. For continuous features indexed by $S_1 = \{k_1, \cdots, k_m\}$, their values in all the samples of $D_t$ should be within a reasonable range, which can be characterized a disjunction of linear inequalities $\Omega = \Omega_1 \vee \cdots \vee \Omega_m$ where $\Omega_i$ denotes a boundary inequality over the feature of the form $l_i \leq x \leq u_i$. For categorical features indexed by $S_2 = \{1, \cdots, d\} \backslash S_1$, we ensure their values fall in the allowable categories $C_1, \cdots, C_{n_l}$ for any feature indexed by $l \in S_2$. By introducing dummy variables $x_{\cdot,l,1}, \cdots, x_{\cdot,l,n_l} \in \{0, 1\}$ where $x_{\cdot,l,j} = \mathbb{I}[x_{\cdot,l} \in C_j], \forall j = 1, \cdots, n_l$ denotes the indicator of whether feature $x_{\cdot,l}$ belongs to a category $C_j$, we impose the categorical constraint $\sum_{j=1}^{n_l} x_{\cdot,l,j} = 1$. Futhermore, strict relationship among features can be expressed in a similar disjunction of inequalities of the form $\sum_{l \in S_1} w_l x_{\cdot,l} + \sum_{l \in S_2} \sum_{j=1}^{n_l} w_{l,j} x_{\cdot,l,j} + b \geq 0$ where $w$ represents the coefficients. As a result, all requirements on feature values and their interdependence can ultimately be formulated as a unified set of logical and linear constraints. Thus, sanity check is reduced to a constraint satisfaction problem and any sample that does not satisfy the predefined constraints are discarded. In practice, thanks to the prior knowledge of LLMs, the generated batch typically satisfies these constraints most of the time. Nevertheless, this check remains indispensable for preventing trivial errors in the generated batch and eliminating semantically meaningless data that could compromise downstream tasks.

**Objective-related cost assessment.** Even if a data batch that passes sanity check, its utility for downstream tasks is not guaranteed, as it may still suffer from issues such as bias, imbalance, noise, etc. Since the generated data ultimately serves a predictive model, additional efforts are needed to ensure models trained on it can achieve high predictive accuracy. Thus, we focus on model-based prediction and evaluate the potential learning objective-related cost. We assume the access to the downstream model $y = f(\boldsymbol{x})$. To facilitate the assessment, we perform bootstrapping from the original set $D_{ori}$ to obtain a dataset $B_t$ of the same size as the generated batch $D_t$, i.e., $n_b$. We combine two sets as $\tilde{D}_t = D_t \bigcup B_t$ and make predictions for all the samples in $\tilde{D}_t$ with the model $f$. Thus, for any sample $\boldsymbol{x}_i$ in $\tilde{D}_t$ with label $y_i$, we have the objective-related cost of a sample for classification tasks:

$$r_i = 1 - p_f(y_i | \boldsymbol{x}_i) \tag{5}$$

where $p_f$ is the predicted probability score given by model $f$. On the one hand, to build a high-quality batch of size $n_b$, we aim to minimize the prediction cost. On the other hand, excessive cost minimization for the model on a small dataset may cause overfitting and potentially degrade the model's generalization performance. Instead of pursuing either extreme, we select the $50\%$ of the $2n_b$ samples in between to build an updated batch. We sort the costs in Eq. (5) and build an empirical distribution of the costs as $F$. Denote $F^{-1}(\beta) \triangleq \inf\{e : F(e) \geq \beta\}$ as the $\beta$-quantile of $\{r_i\}_{i=1}^{2n_b}$. For a specific $\beta \in (0, 0.5)$, we select the samples indexed by $J = \{i_1, \cdots, i_{n_b}\}$ from $\tilde{D}_t$ such that for $\forall i \in J$, we have

$$r_i \in [F^{-1}(\beta), \ F^{-1}(\beta + 0.5)] \tag{6}$$

These selected samples build a refined dataset $\overline{D}_t = \{(\boldsymbol{x}_i, y_i)\}_{i \in J}$, which can be further compared with the bootstrap set $B_t$. We consider the different effects of combining this refined set and the bootstrap set into the existing dataset $D$, which gives two sets: $D_{gen}^1 = D \bigcup \overline{D}_t$ and $D_{gen}^2 = D \bigcup B_t$. Following Mehta et al. (2022); Smith et al. (2023), for any dataset $D'$, the total information gain (IG) of combining $D'$ with $D$ can be calculated by

$$IG(D') = H(D \bigcup D') - H(D) \tag{7}$$

where $H(\cdot)$ is the entropy function. Thus, we can calculate the information gain of the two sets as $IG(D_{gen}^1)$ and $IG(D_{gen}^2)$, and compute their gap as $\Delta = IG(D_{gen}^1) - IG(D_{gen}^2)$, which measures the marginal information gain from the generated data batch $\overline{D}_t$ over the original data batch $\overline{B}_t$ . We only keep the batch of generated data when it gives us enough information, i.e., $\Delta$ is greater than a threshold $\tau$. Thus, if $\Delta > \tau$, we merge the updated batch $\overline{D}_t$ into the existing dataset $D$, otherwise we discard it.

**Diversity inspection.** Beyond validity and objective alignment, it is crucial to ensure that the generated data enhances (at least maintains) the diversity of $D_{ori}$, especially for underrepresented subpopulations. We uncover the underlying structure of $D_{ori}$ by performing clustering. The optimal number of clusters is determined by maximizing the average silhouette score of all the data points in $D_{ori}$. Here, the silhouette score $s(\boldsymbol{x}_i)$ of a data point $\boldsymbol{x}_i$ can be computed by

$$s(\boldsymbol{x}_i) = \frac{z^{coh}(\boldsymbol{x}_i) - z^{sep}(\boldsymbol{x}_i)}{\max\{z^{coh}(\boldsymbol{x}_i), z^{sep}(\boldsymbol{x}_i)\}} \tag{8}$$

where $z^{coh}$ represents the average distance between $\boldsymbol{x}_i$ and all other data points within the same cluster and $z^{sep}$ smallest average distance between $\boldsymbol{x}_i$ and all points in the nearest different cluster. After partitioning $D_{ori}$ into these clusters, we train a multi-class classifier (i.e., an MLP) using the cluster assignments as labels. This classifier effectively learns to map any data point from the feature space to one of the identified data-space regions. If the batch of generated data $D_t$ passes the objective-related cost assessment, we use this trained classifier to predict the cluster for each sample in the batch. To quantify the batch's contribution to overall diversity, we measure the change in the entropy of the cluster label distribution. A batch $D_t$ is accepted only if the percentage entropy improvement of the combined dataset ($D_{ori} \cup D_t$) over the original dataset ($D_{ori}$) exceeds a predefined threshold. This criterion ensures that accepted batches either populate underrepresented clusters or are sufficiently varied to distribute across multiple clusters, thereby expanding coverage without introducing significant skew.

## 3 EXPERIMENTS AND RESULTS

We conduct a comprehensive evaluation of team-then-trim across both simulated studies and real-world applications. In the simulated setting, we recreate common deficiencies in tabular datasets, including imbalance, incompleteness, and noise. For data scarcity, we consider low-data regimes in experiments. For real-world datasets, we examine both the downstream utility of ML models and multiple dimensions of data quality to assess the broader impact of our framework.

### 3.1 SETUP

We employ Llama 3.3 70B Instruct as the backbone LLM for data generation. To validate that our conclusions hold across model families, we also run experiments using Grok 4.1 Fast in the data-imbalance and data-incompleteness settings. For downstream evaluation, we consider: Logistic

Regression, SVM, MLP, and Random Forest. We compare our framework, team-then-trim, against established baselines, including CLLM (Seedat et al., 2024), TVAE (Xu et al., 2019), CTGAN (Xu et al., 2019), and SMOTE (Chawla et al., 2002). For all reported results, except the baseline that uses only $D_{ori}$, the training data consists of the original dataset $D_{ori}$ combined with the generated data produced by team-then-trim or by the baselines. For ablation, we include results from team-then-trim without the QC stage. In the simulated setting, we construct two datasets: (1) a diabetes prediction dataset (Rauba et al., 2024), and (2) a travel behavior dataset (Zhu et al., 2020). For real-world experiments, we adopt datasets: Drug from the UCI repository (Fehrman et al., 2017) and COMPAS from OpenML (Angwin et al., 2016). All reported results are averaged over 10 random seeds corresponding to 10 train-test splits and 4 downstream models. More experimental details are in Appendix C.

## 3.2 SIMULATED STUDIES

**Data imbalance.** In the diabetes dataset, we simulate three levels of risk groups, i.e., low risk (LR), moderate risk (MR), and high risk (HR), with a 7:2:1 ratio to create label imbalance. The size of $D_{ori}$ is 10, following this imbalanced ratio. Table 1 reports the average AUC of different methods. Our team-then-trim framework achieves the strongest and most consistent gains across all groups. Without QC, team-then-trim already enhances AUC in HR group substantially by 12.1% over models trained on $D_{ori}$, highlighting the benefit of LLM-teamed generation in covering rare subpopulations. The consistently superior AUC across classes indicates that our framework with QC is particularly effective in addressing label imbalance in tabular data.

Table 1: Average AUC (%) when the label in the dataset is imbalanced.

| Method | LLM Backbone | LR | MR | HR |
|---|---|---|---|---|
| $D_{ori}$ | - | 63.92 | 59.16 | 64.24 |
| SMOTE | - | 68.99 | 58.73 | 68.66 |
| TVAE | - | 68.79 | 60.75 | 65.22 |
| CTGAN | - | 67.30 | 59.61 | 67.00 |
| CLLM | Grok 4.1 Fast | 64.29 | 54.72 | 66.87 |
| CLLM | Llama 3.3 | 64.43 | 57.15 | 65.79 |
| Team-then-Trim (w/o QC) | Grok 4.1 Fast | 65.89 | 56.64 | 75.90 |
| Team-then-Trim | Grok 4.1 Fast | 66.04 | 56.67 | 76.45 |
| Team-then-Trim (w/o QC) | Llama 3.3 | 65.57 | 57.16 | 76.34 |
| Team-then-Trim | Llama 3.3 | **71.55** | **62.03** | **76.98** |

**Data incompleteness.** To simulate real-world incompleteness, we construct $D_{ori}$ with LR:MR:HR = 8:2:0, such that the HR group constitutes a missing subpopulation in the training data. The size of $D_{ori}$ is set to 10. The test set retains the distribution with LR:MR:HR = 7:2:1. Table 2 presents average AUC under this setting. Classical generation baselines completely fail to recover the missing HR subgroup. Although CLLM leverages the LLM for generation and attains relatively high AUC in the MR group, it generalizes poorly to the HR group, indicating that without explicit LLM teaming and rigorous QC, the generated samples are often noisy or misaligned. By contrast, team-then-trim demonstrates strong performance of 14.18% gain over CLLM in the HR group.

Table 2: Average AUC (%) when the dataset is incomplete in a subpopulation.

| Method | LLM Backbone | LR | MR | HR |
|---|---|---|---|---|
| $D_{ori}$ | - | 58.59 | 48.69 | / |
| SMOTE | - | 64.70 | 48.52 | / |
| TVAE | - | 62.81 | 49.85 | / |
| CTGAN | - | 65.14 | 49.87 | / |
| CLLM | Grok 4.1 Fast | 82.58 | 48.99 | 53.68 |
| CLLM | Llama 3.3 | 63.68 | **54.47** | 50.49 |
| Team-then-Trim (w/o QC) | Grok 4.1 Fast | 82.44 | 51.70 | 56.50 |
| Team-then-Trim | Grok 4.1 Fast | **82.98** | 52.59 | 58.13 |
| Team-then-Trim (w/o QC) | Llama 3.3 | 64.76 | 52.30 | 62.34 |
| Team-then-Trim | Llama 3.3 | 71.16 | 52.26 | **64.67** |

**Data noise.** To simulate data noise, we encode the outcome variables in both datasets as binary indicators. Larger flip ratios of the true labels correspond to higher levels of label corruption, thereby creating increasingly noisy datasets. In the diabetes dataset, at a low noise level (flip ratio = 0.2, shown in Fig. 3a), team-then-trim outperforms all models when the number of original samples is small. As the number of original samples increases, the clean data alone already yields high AUC due to the mild noise. As noise increases to 0.3 (shown in Fig. 3b) and 0.4 (shown in Fig. 3c), team-then-trim sustains stable gains. This demonstrates the framework's robustness to label corruption

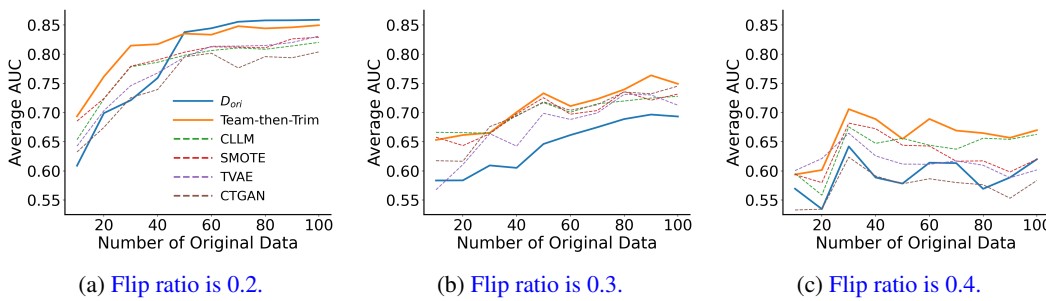

(a) Flip ratio is 0.2.   (b) Flip ratio is 0.3.   (c) Flip ratio is 0.4.

Figure 3: Average AUC across different sizes of $D_{ori}$ under flip ratios of {0.2, 0.3, 0.4} on Diabetes dataset.

even under moderate to severe noise. Additional performance metrics on diabetes data are presented in Appendix Table 7 and t-SNE visualizations are in Appendix Fig. 9.

Table 3 shows the average AUC under no flip, flip ratios of 0.3, and 0.4 on the TravelBehavior dataset, with 60 original samples. The observed trends mirror those in the diabetes dataset. Overall, team-then-trim enhances downstream utility across noisy settings in both datasets, with particularly strong gains under higher noise levels, highlighting its effectiveness as a generative framework for challenging real-world tabular data.

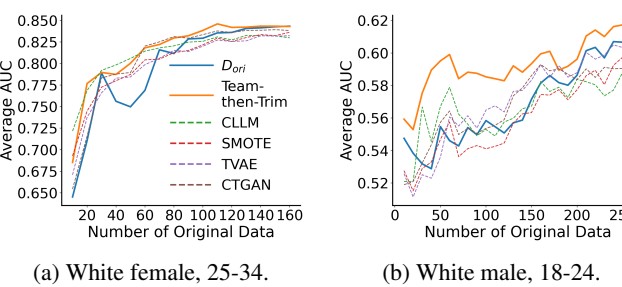

(a) White female, 25-34.   (b) White male, 18-24.

Figure 4: Average AUC for two demographic groups on Drug dataset.

Table 3: Average AUC (%) under flip ratios of {0, 0.3, 0.4} on TravelBehavior Dataset.

| Method | No Flip | 0.3 | 0.4 |
|---|---|---|---|
| $D_{ori}$ | **98.55** | 60.61 | 50.02 |
| SMOTE | 98.54 | 58.95 | 51.03 |
| TVAE | 96.62 | 57.80 | 49.11 |
| CTGAN | 91.40 | 58.91 | 49.26 |
| CLLM | 97.73 | 62.05 | 53.14 |
| Team-then-Trim (w/o QC) | 98.08 | 62.49 | 53.02 |
| Team-then-Trim | 98.24 | **63.44** | **53.38** |

## 3.3 REAL-WORLD DATA

### 3.3.1 DOWNSTREAM UTILITY

To assess model performance across diverse populations, we partition the Drug dataset into two subgroup datasets Drug-A and Drug-B based on their demographic features (age, gender, and ethnicity). Fig. 4a corresponds to the subgroup whose age is between 25 and 34, female, and White, while Fig. 4b focuses on White male with age between 18 and 24. Across both groups, team-then-trim outperforms baseline methods, particularly in low- and medium-data regimes. For the 18-24 male subgroup, team-then-trim delivers the highest AUC across all sample sizes from 10 to 250. The larger improvement of team-then-trim in the 18-24 male subgroup is attributed to the binary label imbalance, where 74.07% of samples are labeled as 1, compared to 63.11% in the 25-34 female subgroup. Overall, our method enhances model utility in both balanced and imbalanced settings.

Table 4: Average downstream utility (%) across different sizes of $D_{ori}$ (10 to 200) on COMPAS dataset.

| Method | Accuracy | AUC | F1 | Recall |
|---|---|---|---|---|
| $D_{ori}$ | 60.89 | 64.12 | 58.69 | 60.00 |
| SMOTE | 60.77 | 64.86 | 58.43 | 60.20 |
| TVAE | 60.81 | 64.32 | 58.63 | 60.62 |
| CTGAN | 60.54 | 64.61 | 58.23 | 60.25 |
| CLLM | 61.70 | 65.54 | 59.26 | 60.90 |
| Team-then-Trim (w/o QC) | 61.32 | 66.82 | 61.46 | **66.87** |
| Team-then-Trim | **62.05** | **67.29** | **61.84** | 66.11 |

Table 5: Average data quality (%) across different sizes of $D_{ori}$ on real-world data.

| Dataset | Method | Detection | $\alpha$-Precision | $\beta$-Recall |
|---|---|---|---|---|
| DRUG-A | SMOTE | 65.71 | 53.97 | 35.50 |
| | TVAE | 65.59 | 53.89 | 35.73 |
| | CTGAN | 64.60 | 54.01 | 35.77 |
| | CLLM | 63.27 | 58.12 | 37.25 |
| | Team-then-Trim (w/o QC) | 63.00 | 87.38 | 42.62 |
| | Team-then-Trim | **56.68** | **88.76** | **48.45** |
| DRUG-B | SMOTE | 66.30 | 51.20 | 34.01 |
| | TVAE | 65.99 | 51.12 | 33.34 |
| | CTGAN | 66.15 | 51.23 | 33.25 |
| | CLLM | 63.67 | 56.53 | 35.47 |
| | Team-then-Trim (w/o QC) | 64.91 | 89.70 | 44.40 |
| | Team-then-Trim | **57.81** | **90.96** | **49.29** |
| COMPAS | SMOTE | 52.91 | 88.47 | 46.52 |
| | TVAE | 51.37 | 89.94 | 50.32 |
| | CTGAN | 51.12 | 91.23 | 51.03 |
| | CLLM | 50.26 | **91.58** | 48.63 |
| | Team-then-Trim (w/o QC) | 52.29 | 87.65 | 47.55 |
| | Team-then-Trim | **49.17** | 91.11 | **51.29** |

Table 4 provides a summary of downstream utility across varying sizes of $D_{ori}$ on COMPAS dataset. Team-then-trim achieves the best overall performance across all four metrics. Notably, the ablation team-then-trim w/o QC also shows competitive improvements (e.g., recall of 66.87%), confirming that LLM teaming alone provides value. Also, the full QC pipeline consistently yields higher accuracy, AUC and F1 scores, demonstrating that rigorous trimming enhances the informativeness and task-alignment of generated samples. Detailed results on the average AUC of four ML models are shown in Appendix Fig. 10.

### 3.3.2 DATA QUALITY

We evaluate the quality of generated data using three metrics: detection (Liu et al., 2023), $\alpha$-precision (Alaa et al., 2022), and $\beta$-recall (Alaa et al., 2022). Detection measures whether synthetic data can be distinguished from real data, with lower scores indicating higher similarity (thus better quality). $\alpha$-precision quantifies the fidelity of generated samples, while $\beta$-recall captures their diversity. More descriptions on evaluation metrics are provided in the Appendix C. Table 5 shows the data quality of different generation methods on the DRUG-A (subgroup in Fig. 4a), DRUG-B (subgroup in Fig. 4b), and COMPAS datasets. Team-then-trim consistently outperforms baseline methods except $\alpha$-precision in the COMPAS dataset. This is because $\alpha$-precision emphasizes high-fidelity reconstruction of the densest regions of the real distribution, whereas our framework intentionally trades a small amount of local fidelity for improved coverage and objective alignment. The QC pipeline is designed to admit batches that increase information gain and expand cluster-level diversity, which naturally encourages exploration beyond the tight, high-probability core modes favored by $\alpha$-precision. COMPAS is a dataset with well-documented fairness issues (Wang et al., 2019), which means its real distribution is highly concentrated and exhibits limited feature variability. Our diversity-oriented trimming can slightly pull samples away from the exact support of the majority mode, yielding marginally lower $\alpha$-precision. However, this trade-off is beneficial for downstream learning, as reflected in consistently stronger $\beta$-recall, detection scores, and predictive performance. These results demonstrate that team-then-trim generates synthetic data that are simultaneously indistinguishable from real data, faithful to the real distribution, and diverse enough to cover rare or missing modes. The consistent improvements across heterogeneous datasets reinforce the effectiveness of LLM teaming coupled with rigorous QC in producing high-quality tabular data.

## 4 CONCLUSION

In this work, we introduced team-then-trim, a novel assembly-line framework for tabular data generation that leverages the complementary strengths of LLMs and a principled QC pipeline. By decomposing generation into specialized LLM teams and rigorously trimming outputs through three-stage

checks, our approach systematically transforms raw generations into high-quality synthetic datasets. Extensive experiments across simulated deficiencies, including imbalance, incompleteness, noise and scarcity, and real-world applications demonstrate that team-then-trim consistently outperforms state-of-the-art generation methods. Notably, the framework not only improves downstream utility across diverse classifiers but also achieves superior fidelity, diversity, and indistinguishability from real data. These results highlight the potential of structured LLM teaming, coupled with rigorous data QC, to bridge critical gaps in learning scenarios with data deficiencies.

## ETHICS AND REPRODUCIBILITY STATEMENT

**Ethics Statement.** This work focuses on developing a framework for generating synthetic tabular data to address common deficiencies such as imbalance, incompleteness, noise and scarcity in datasets. All datasets used in this paper are either simulated or publicly available datasets. No private, personally identifiable, or otherwise sensitive data were used. We adhered to responsible research practices throughout this work.

**Reproducibility Statement.** Detailed instructions for dataset access, simulation parameters, preprocessing steps and methods implementation are provided in Section 3 and Appendix C. Our code will be released publicly upon acceptance.

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

## A    RELATED WORK

**Tabular Data Generation.**    Synthetic tabular data generation has evolved from statistical methods to deep generative models (Chawla et al., 2002; Xu et al., 2019; Goyal & Mahmoud, 2024; Dankar & Ibrahim, 2021). GOGGLE (Liu et al., 2023) learns an explicit relational structure among columns and jointly trains a message-passing VAE, enabling better modeling of sparse, heterogeneous dependencies in tabular data. TabDiff (Shi et al., 2024) proposes a unified mixed-type diffusion model that jointly handles numerical and categorical columns in continuous time, with feature-wise learnable noise schedules and a transformer denoiser. LLMs represent a paradigm shift, leveraging vast pre-trained knowledge to generate data for scenarios absent in the original dataset (Tang et al., 2023; Patel et al., 2024; Shimabucoro et al., 2024). CLLM (Seedat et al., 2024) leverages a frozen LLM to generate tabular samples in ultra-low-data settings and then curates them via learning-dynamics signals. DataEnvGym (Khan et al., 2024) introduces a modular teacher-student testbed where data generation agents plan and synthesize training data in a feedback loop to improve a student model, framing generation as sequential decision-making. Goyal & Mahmoud (2025) present a practical platform that fine-tunes LLMs and integrates differential privacy to generate datasets while preserving sensitive information. Yang et al. (2024) define table similarity via realistic analyst-style transformations and introduces an LLM-driven pipeline that generates large-scale pairs of similar tables to train and evaluate table-level embeddings. However, single LLM can hallucinate unseen categories, violate hard constraints and struggle to capture complex conditional distributions. Our framework, team-then-trim, addresses these gaps by decomposing generation into a manufacturing assembly line and integrating a rigorous three-stage QC pipeline to ensure the final dataset is plausible and task-aligned.

**Data Quality Control.**    QC for synthetic data has matured from ad-hoc checks to systematic, scalable pipelines that validate schemas and repair errors (Baur et al., 2020; Tamm & Nikiforova, 2025; Alaa et al., 2022). Schelter et al. (2018) supports incremental metric computation on growing datasets, and adds ML-assisted predictability and anomaly detection to automate large-scale data quality verification. Building on this work, Schelter et al. (2019) propose a differential extension that represents data-quality metrics as algebraic states with commutative-monoid properties, enabling incremental, partition-aware verification without rescanning previously processed data. Recent advances increasingly emphasize QC for LLM-generated data (Chen et al., 2024a; Hu et al., 2025). Sousa et al. (2024) apply a human protocol to deduplicate, filter out-of-scope samples before using the curated set from LLMs and underscore the need for human validation. Wang et al. (2023) introduce a model-agnostic, differential privacy-preserving post-processing method that reweights a synthetic dataset via information projection so that selected utility measures, e.g., moments, correlations, match noisy targets from the real data. LLM-TabLogic (Long et al., 2025) uses LLM prompting to infer and compress inter-column logical relationships and then conditions a latent diffusion generator on these constraints, producing synthetic tables that better preserve logical consistency while maintaining strong fidelity and privacy. CROWDSELECT (Li et al., 2025) aggregates multiple LLMs' responses and reward scores to compute three metrics with diversity clustering and multi-metric normalization to select synthetic instruction data. However, these methods remain pointwise and model-agnostic, and do not guarantee that admitted samples are useful for the downstream task or that they cover rare modes. Our framework, team-then-trim, addresses these gaps by coupling a manufacturing-style assembly line with a three-stage QC that is explicitly task-linked and batch-level.

## B    PROMPT EXAMPLES

We provide prompt examples of the LLM task manager in Fig. 6, along with one of its assigned roles, the LLM Demographic and Lifestyle Synthesizer, in Fig. 7, and a subsequent role, the LLM Glucose Regulation Simulator, in Fig. 8. In the prompts, we provide detailed task instructions along with the definitions of all features to help LLMs clearly understand the assignment and generation tasks.

| Age | Physical Activity | BMI | Blood Pressure | Cholesterol | HbA1c | Fasting Glucose | Insulin | y |
|---|---|---|---|---|---|---|---|---|
| 32 | 2 | 24 | 109 | 222 | 5.26 | 104 | 8 | 1 |
| 38 | 2 | 30 | 94 | 129 | 5.5 | 85 | 27 | 2 |
| 56 | 5 | 20 | 127 | 153 | 4.87 | 115 | 15 | 1 |
| 44 | 4 | 16 | 93 | 163 | 5.15 | 130 | 94 | 3 |
| ... | | | | | | | | |

Figure 5: Examples of original data in Diabetes dataset.

## C  EXPERIMENTAL DETAILS

**Datasets simulation.**    In the simulated studies, the features of the Diabetes dataset and their co-efficients are generated following the distributions in Rauba et al. (2024), with label probabilities computed using the sigmoid function. Figure 5 shows examples of original data in the simulated Diabetes. The dataset includes eight features and one outcome label $y$. For the TravelBehavior dataset, features are simulated according to the distributions in Feng et al. (2020); Lin et al. (2024), and ground-truth labels are assigned using their proposed Latent Decision Threshold model. The simulated TravelBehavior dataset consists of four features and one binary outcome label. In both datasets, we construct a test set of 500 samples, while the size of the original training data varies from 10 to 100, increasing in steps of 10. Unless otherwise specified, the data generation adheres to the aforementioned distributions. We use symmetric label flipping in all noise experiments. Specifically, for a given flip ratio, we uniformly sample a subset of data points without conditioning on class and randomly flip their labels. Thus, every class is equally likely to be corrupted, and no class receives preferential or targeted noise.

**Real-world datasets.**    For the real-world applications, the Drug dataset from the UCI repository (Fehrman et al., 2017) is partitioned by age, gender, and ethnicity, and we only retain subgroups with more than 250 observations to ensure sufficient training and test data. After partition, the dataset consists of 24 features and one binary label outcome. For each subgroup, 100 samples are held out as the test set, with the remainder used for training after stratifying by label distribution. In the COMPAS dataset from OpenML (Angwin et al., 2016), the number of training data varies across experiments with balanced labels, and 500 samples are reserved for testing. The COMPAS dataset consists of 13 features and one binary label outcome.

**Data generation.**    Considering plausibility of evaluation and the token limits, we generate 10 data points in each batch for Diabetes and Drug datasets, and 20 data points in each batch for the TravelBehavior and COMPAS datasets. In the data imbalance and incompleteness experiments, the number of generated batches before trimming is fixed at 10. In all other experiments with varying original data, it is set equal to the size of the original dataset; for example, with 10 original samples, one batch is generated. No additional batches are added after trimming. This design ensures that the amount of synthetic data scales proportionally with the available data, preventing over-generation.

**QC.**    In step 2 objective-related cost assessment of QC pipeline, to calculate the threshold, we run 1000 times to obtain the mean and standard deviation. The coefficient of standard deviation ranging from 0 to 3 is selected by 5-fold cross validation. In step 3 diversity-related monitoring, we choose the number of clusters from $\{3, 4, 5\}$ to maximize the silhouette scores. The entropy improvement threshold is set as $30\%$.

**Models.**    All the downstream models are implemented by scikit-learn (Pedregosa et al., 2011). All the experiments are run on Apple M3 Pro. We use SynthCity (Qian et al., 2023) to implement the baseline methods and to evaluate the $\alpha$-precision and $\beta$-recall, except for CLLM, which follows the implementation details in Seedat et al. (2024). The detection score is computed following the procedure in Liu et al. (2023).

**Prompt Example for LLM Task Manager**

You are a Task Manager in a diabetes research system, overseeing the assignment of roles to various Large Language Models (LLMs). Each LLM is to be assigned a specific role, such as "Alternative Designer" or "Simulated User," to augment distinct aspects of dataset concerning individual health condition. The dataset includes individual features with their meanings provided and the outcome label is y, the diabetes indicator .

**Task:** Assign appropriate roles to LLMs and organize them into a sequence that facilitates collaborative augmentation of the dataset. This sequence should reflect the logical relationships and dependencies between the roles to maximize the efficiency and effectiveness of data augmentation.

**Features and Their Meanings:**
HbA1c: Hemoglobin A1c levels
FastingGlucose: Fasting glucose levels
Age: Age
BMI: Body Mass Index
BloodPressure: Blood pressure
Cholesterol: Cholesterol levels
Insulin: Insulin levels
PhysicalActivity: Physical activity levels
y: Diabetes indicator (0 = No diabetes, 1 = Diabetes)

**Output Requirements:** Generate a structured JSON detailing the role assignments and the sequence of augmentation. You are encouraged to create new roles or modify existing ones as necessary. Ensure that each feature is managed by one specific role. Also, organize the roles in a sequence ("Relationship") where the output of one role feeds into the next, facilitating seamless data augmentation. Here is the required JSON structure:
{"Roles":
{"Role1": {"Name": "Role name", "Features": "Features it needs to augment"},
"Role2": {"Name": "Role name", "Features": "Features it needs to augment"},
"Role3": {"Name": "Role name", "Features": "Features it needs to augment"},
"Role4": {"Name": "Role name", "Features": "Features it needs to augment"}},
"Relationship":
{"Order 1": "Which LLM should first augment features?",
"Order 2": "Which LLM should secondly augment features?",
"Order 3": "Which LLM should thirdly augment features?",
"Order 4": "Which LLM should lastly augment features?"}}.

Figure 6: Prompt example of task manager LLM.

---

**Prompt Example for LLM Demographic and Lifestyle Synthesizer**

You are a Demographic and Lifestyle Synthesizer in diabetes research system. Your task is to thoughtfully and realistically generate 10 novel individuals. Please think step by step and incorporate patterns learned from real-world users. You need to generate diverse and plausible combinations of the following features: "Age", and "PhysicalActivity". You are provided with 10 representative examples of real users, which include both these features and additional ones known to be associated with decision making. Use these examples to infer realistic feature relationships, but do not copy them directly.

**Features and Their Meanings:**
Age: Age
PhysicalActivity: Physical activity levels
BMI: Body Mass Index
BloodPressure: systolic blood pressure
Cholesterol: Cholesterol levels
HbA1c: Hemoglobin A1c levels
FastingGlucose: Fasting glucose levels
Insulin: Insulin levels
y: Diabetes indicator (0 = No diabetes, 1 = Diabetes)

**Example data:**
{example_data_text}

**Output Requirements:** The output should be a markdown code snippet formatted in the following schema, including the leading and trailing "json" and "```":
```json
{{
"Age": string,
"PhysicalActivity": string.
}}
```

Figure 7: Prompt example of an assigned role by the LLM, i.e., demographic and lifestyle synthesizer.

**Evaluation metrics.** We use various evaluation metrics to capture different aspects of downstream predictive performance and quality of generated data. AUC measures the probability that a randomly chosen positive sample receives a higher predicted score than a randomly chosen negative sample. AUC is threshold-independent and particularly informative in imbalanced settings, which is why it serves as one of our primary metrics. Accuracy provides a holistic overview of performance but may be less informative under label imbalance. Recall measures the fraction of positive samples that are correctly identified. F1-score balances the trade-off between precision and recall and serves as a compact summary of classifier performance when neither error mode is dominant.

## D ADDITIONAL RESULTS

**LLM workers.** Task manager LLM assigns different roles for different datasets. For diabetes dataset, the LLM roles are Demographic and Lifestyle Synthesizer, Glucose Regulation Simulator, Cardiometabolic Generator, and Diabetes Expert. For TravelBehavior dataset, the LLM roles are Alternative Designer, Incentive Allocator, and Decision Predictor. For Drug dataset, the LLM roles are Demographic Profiler, Personality Synthesizer, Drug Usage Generator, and Nicotine Propensity Estimator. For COMPAS dataset, the LLM roles are Demographic Profile Designer, Juvenile History Synthesizer, Criminal Record Analyst and Recidivism Outcome Evaluator.

**Data incompleteness.** We visualize the t-SNE embeddings of the MR and HR groups in Fig. 9 given the training set missing the HR subgroup on diabetes data. In both groups, the team-then-trim samples closely intermingle with the test samples, indicating that the generated data preserves the manifold structure of the real-world distribution. This alignment reflects the ability of our LLM-teamed generation, followed by QC, to enrich existing but underrepresented or even missing sub-populations with high fidelity. The results show that, unlike classical baselines, team-then-trim can generalize beyond observed data to reconstruct missing modes in the distribution. This capability enables downstream models to achieve strong performance on previously unseen subgroups.

---

**Prompt Example for LLM Glucose Regulation Simulator**

You are a Glucose Regulation Simulator in a diabetes research system. Your task is to thoughtfully and realistically generate 10 novel individuals given their features Age and PhysicalActivity. Please think step by step and incorporate patterns learned from real-world users. You need to generate diverse and plausible combinations of the following features: "HbA1c", "FastingGlucose", and "Insulin". You are provided with 10 representative examples of real users, which include both these features and additional ones known to be associated with decision making. Use these examples to infer realistic feature relationships, but do not copy them directly

**Features and Their Meanings:**
Age: Age
PhysicalActivity: Physical activity levels
BMI: Body Mass Index
BloodPressure: systolic blood pressure
Cholesterol: Cholesterol levels
HbA1c: Hemoglobin A1c levels
FastingGlucose: Fasting glucose levels
Insulin: Insulin levels
y: Diabetes indicator (0 = No diabetes, 1 = Diabetes)

**Example data:**
{example_data_text}

**Output Requirements:** The output should be a markdown code snippet formatted in the following schema, including the leading and trailing "json" and "```":
```json
{
{"Age": 23,
"PhysicalActivity": 4,
"HbA1c": string,
"FastingGlucose": string,
"Insulin": string.
},
{"Age": 47,
"PhysicalActivity": 2,
"HbA1c": string,
"FastingGlucose": string,
"Insulin": string.
},
...
}
```

Figure 8: Prompt example of an assigned role by the LLM, i.e., glucose regulation simulator, following LLM demographic and lifestyle synthesizer.

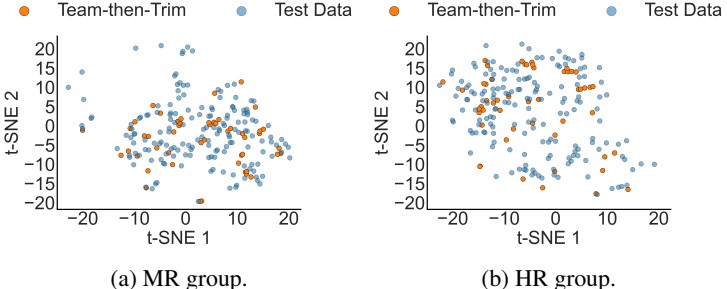

(a) MR group.        (b) HR group.

Figure 9: t-SNE plots of MR and HR groups when the dataset is incomplete.

Table 6: Average AUC (%) with 95% CI when the label in the dataset is imbalanced on simulated Diabetes dataset.

| Methods | LR | MR | HR |
|---|---|---|---|
| $D_{ori}$ | $0.64 \pm 0.15$ | $0.59 \pm 0.14$ | $0.64 \pm 0.04$ |
| CLLM | $0.64 \pm 0.12$ | $0.57 \pm 0.10$ | $0.66 \pm 0.16$ |
| Team-then-Trim (w/o QC) | $0.66 \pm 0.11$ | $0.57 \pm 0.09$ | $0.76 \pm 0.09$ |
| Team-then-Trim (after QC Step 1) | $0.66 \pm 0.11$ | $0.57 \pm 0.09$ | $0.76 \pm 0.09$ |
| Team-then-Trim (after QC Step 2) | $0.68 \pm 0.14$ | $0.59 \pm 0.12$ | $0.76 \pm 0.10$ |
| Team-then-Trim | $\mathbf{0.72 \pm 0.19}$ | $\mathbf{0.62 \pm 0.17}$ | $\mathbf{0.77 \pm 0.11}$ |

**Analysis of QC steps.** Table 6 shows that each QC stage incrementally improves AUC under the data-imbalance setting of simulated Diabetes dataset. The LLM teaming output (w/o QC) already enhances HR performance by expanding coverage of rare cases, but offers limited gains for LR and MR due to noisy or misaligned samples. Sanity check (Step 1) yields no change, consistent with the fact that LLM teaming rarely violates basic constraints. Objective-related cost assessment (Step 2) produces the first clear improvement by removing samples with poor predictive alignment, which increases AUC in both LR and MR. Diversity-related monitoring (Step 3) delivers the final boost by admitting only batches that enhance cluster-level coverage. Together, the three stages refine raw generations into data that is simultaneously valid, task-aligned, and diverse, yielding the highest AUC across all groups. The confidence intervals (CIs) become slightly wider after QC because the trimming process reduces the number of retained samples and increases the variability across random seeds.

**Data noise.** Table 7 reports the average performance on the diabetes dataset under varying levels of label noise, with flip ratios of 0.2, 0.3, and 0.4. Under moderate and high label corruption, both team-then-trim and its ablation without QC sustain strong performance across the metrics, delivering the most consistent improvements. The gains over $D_{ori}$ and all baselines highlight the effectiveness of combining LLM teaming with rigorous trimming in filtering out mislabeled or uninformative samples. At a low noise level (flip ratio = 0.2), all methods achieve relatively high AUC, F1 scores and recall, reflecting the mild corruption. Team-then-trim achieves the best AUC of 81.42% and competitive F1 and recall. This indicates that even when data are only lightly corrupted, our framework provides additional robustness without overfitting to mislabeled samples.

**Computational costs of LLM-based data generation.** To contextualize the computational costs of our framework, we examine the token usage and runtime of each LLM component during data generation under the diabetes data-imbalance setting. Because the QC stage relies on lightweight procedures and small models relative to our data size, its computational overhead is negligible compared with the cost of LLM-based generation. The final generated dataset has a dimensionality of 10 rows × 9 columns. As shown in Table 8, team-then-trim incurs moderately higher token consumption and runtime than CLLM; however, the framework remains cost-efficient given its substantially stronger downstream utility. Specifically, Table 1 shows that team-then-trim improves AUC by 7.12% in the LR group, 4.88% in the MR group, and 11.19% in the HR group compared with CLLM. These gains highlight that structured LLM teaming does not introduce prohibitive overhead and, in fact, achieves a highly favorable cost–benefit tradeoff in data-deficiency regimes.

**Model utility.** Fig. 10 presents the average AUC of four ML models trained on the augmented COMPAS dataset across varying sizes of original data from 10 to 200. Across all models, team-then-trim outperforms baseline approaches under most sizes of original data. Importantly, while classical baselines occasionally achieve moderate gains, their improvements are inconsistent and often fall short of the stability achieved by team-then-trim. These trends indicate that the coordinated LLM-teaming plus trimming pipeline enhances model utility in ways that benefit a broad spectrum of downstream learners.

Table 7: Average performance (%) across different sizes of $D_{ori}$ (from 10 to 100) when data is noisy in the diabetes data.

| Flip Ratio | Method | AUC | F1 | Recall |
|---|---|---|---|---|
| | $D_{ori}$ | 79.00 | **73.26** | 75.13 |
| | SMOTE | 78.73 | 70.81 | **75.90** |
| | TVAE | 77.47 | 72.32 | 75.35 |
| 0.2 | CTGAN | 75.38 | 72.99 | 74.07 |
| | CLLM | 77.99 | 68.70 | 68.91 |
| | Team-then-Trim w/o QC | 80.83 | 71.10 | 72.39 |
| | Team-then-Trim | **81.42** | 72.62 | 74.84 |
| | $D_{ori}$ | 64.42 | 64.66 | 67.41 |
| | SMOTE | 69.77 | 63.03 | 69.57 |
| | TVAE | 63.67 | 64.76 | **70.21** |
| 0.3 | CTGAN | 66.44 | 62.87 | 68.65 |
| | CLLM | 69.96 | 61.16 | 61.88 |
| | Team-then-Trim w/o QC | 70.99 | 63.41 | 65.69 |
| | Team-then-Trim | **71.84** | **64.78** | 67.69 |
| | $D_{ori}$ | 59.18 | 58.34 | 58.91 |
| | SMOTE | 62.66 | 59.17 | 59.53 |
| | TVAE | 61.52 | 56.72 | 57.82 |
| 0.4 | CTGAN | 57.40 | 54.20 | 55.23 |
| | CLLM | 63.86 | **60.64** | 49.73 |
| | Team-then-Trim w/o QC | **66.61** | 58.12 | 58.48 |
| | Team-then-Trim | 65.94 | 58.95 | **59.76** |

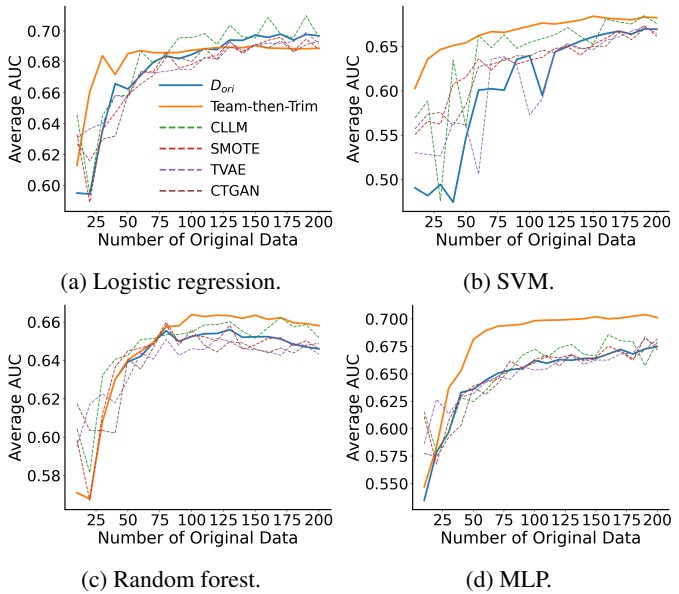

(a) Logistic regression.

(b) SVM.

(c) Random forest.

(d) MLP.

Figure 10: Average AUC of four ML models across original data from 10 to 200 on COMPAS dataset.

Table 8: Token usage and runtime cost of LLM-based data generation under the Diabetes (Imbalanced) setting.

| Method | Components | Tokens (Input) | Tokens (Output) | Time (s) |
|---|---|---|---|---|
| CLLM | – | 771 | 872 | 6.56 |
| Team-then-Trim | LLM Worker 1 | 832 | 387 | 4.83 |
| | LLM Worker 2 | 1831 | 887 | 6.08 |
| | LLM Worker 3 | 2457 | 1483 | 7.16 |
| | LLM Worker 4 | 2533 | 1507 | 7.46 |

## E  EXAMPLE: THE ADVANTAGE OF LLM TEAMING

Fig. 11 presents generated data examples from a single LLM and from LLM teaming on COMPAS dataset. The feature "juv_fel_count" denotes the number of juvenile felonies of a defendant, while "priors_count" represents the total number of prior criminal records. By definition, "priors_count" should always be greater than or equal to "juv_fel_count". As shown in Fig. 11a, data generated by a single LLM fails to respect this rule, whereas in Fig. 11b, our proposed LLM teaming framework successfully produces data that adheres to such basic consistency constraints.

| juv_fel_count | juv_misd_count | juv_other_count | priors_count | juv_fel_count | juv_misd_count | juv_other_count | priors_count |
|---|---|---|---|---|---|---|---|
| 2 | 0 | 0 | 0 | 1 | 0 | 1 | 2 |
| 3 | 0 | 1 | 2 | 0 | 1 | 3 | 4 |
| ... | | | | ... | | | |

(a) Single LLM.                     (b) LLM teaming.

Figure 11: Comparison of generated data on COMPAS from a single LLM and the proposed LLM teaming framework.

## F  LLM USAGE

We employ LLMs to generate tabular data as part of our framework. Beyond this, all written content in this paper is authored by us. After completing the draft, we use LLMs to assist with grammar checking.

