# OpenReview forum: "Team, Then Trim: An Assembly-Line LLM Framework for High-Quality Tabular Data Generation"
_ICLR.cc/2026/Conference — Submitted to ICLR 2026_

### Official Review · Reviewer_fpt4 · 2025-10-27

**Soundness:** 3
**Presentation:** 3
**Contribution:** 1
**Rating:** 4
**Confidence:** 5

**Summary:**

The paper proposes the team-then-trim framework for tabular data generation in low-data regimes (n<100).

It’s two parts (1) Multi-agent synthetic data generation & (2) QC pipeline.

Assessed on different tabular data settings vs other generative and LLM based synthetic data generators.

Core contribution: LLM decomposition via multi-agent workers and a different curation mechanism (not the idea of the generation+curation pipeline itself, which prior work proposed)

**Strengths:**

- Significance: Tackles an important and well-studied problem with high impact in many domains

- Originality: Proposes an interesting idea of multi-agent synthetic data generation + a multi-step QC pipeline.
The idea of feature decomposition and the generation to respect dependencies is great.

- Quality: Good set of experiments in many scenarios: (imbalance, incompleteness, noise, scarcity) + multiple downstream models.
Seems to outperform existing methods on the settings tested (albeit minimally)

- Clarity: clearly written paper

**Weaknesses:**

(1) Limited novelty: basically the same idea as CLLM, just with a multi-agent approach + different QC approach. Better positioning is needed to understand the gain because there are more LLM calls (via more agents)

(2) Inconsistent results: while train-then-trim mostly outperforms other approaches, it is not universally the case. Understanding when it helps and when it doesn’t and why is important

(3) Missing computational cost: given the extra LLM calls via multi-agent, it is important to understand the performance cost vs performance gain trade-off

(4) Source of gain: It’s useful to understand if the source of gain is from the multi-agent generation or the QC approach. What if the CLLM curation mechanism were applied to the multi-agent generation, how would it perform with the new QC mechanism? i.e. an important ablation is Train-then-team + CLLM curation, CLLM generation + new QC mechanism

(5) Analysis of teaming — more in-depth analysis on when teaming provides value. What types of datasets, what dataset sizes, when should it be used and when does it provide minimal value

(6) Additional LLMs: the paper only uses Llama as the backbone LLM. It is important to try different architectures of LLMs, different sizes and more recent LLMs, such that we know as of today if multi-agent is needed and for all LLMs.

**Questions:**

- Please can you had the computational costs and number of LLM calls

- Please can you add analysis with other LLMs bases, sizes/parameters (and generally more recent)

- Please can you add this abaltion to understand which component (generation vs curation) drives improvement? Team-then-Trim generation + CLLM's learning dynamics curation vs CLLM's single-LLM generation + Team-then-Trim's 3-stage QC

- Please can you add std dev for all results over the 10 seeds

- The datasets used are public and might be known to the LLM. Some analysis needs to be done on newer/private datasets to assess generalisation

---

> ### Author Response · Authors · 2025-11-30
> **Official Rebuttal to Reviewer fpt4 (1/3)**
>
> Thank you for your valuable and insightful comments. Please see our response below.
>
> >(1) Limited novelty: basically the same idea as CLLM, just with a multi-agent approach + different QC approach. Better positioning is needed to understand the gain because there are more LLM calls (via more agents)
>
> While both team-then-trim and CLLM leverage LLMs for tabular data generation based on small data followed by a post-generation refinement process, they follow different motivations and designs. CLLM focuses on individually curating synthetic samples given a small dataset by performing single-LLM generation followed by sample-wise curation. In contrast, the central objective of our Team-then-trim is to systematically enrich the data and solve dataset deficiencies, e.g., imbalance, selection bias, logical inconsistencies, and missing subpopulation coverage. Our method design is driven by this objective. Team-then-trim introduces a fundamentally different generation paradigm, i.e., structured, multi-agent, dependency-aware data manufacturing, supported by a batch-wise, multi-objective QC pipeline. CLLM treats the LLM as a single, unstructured generator: one LLM receives a flat prompt, emits full samples i.i.d., and a classifier curates them. LLM teaming in our team-then-trim enables inter-feature constraints to be preserved during generation rather than only relying on post-hoc repair. The increased number of LLM calls is justified because it expands capability, not redundancy. Figure 10 shows that replacing the multi-agent scheme with a single LLM (CLLM-style) cannot capture structured dependencies, demonstrating that the additional calls serve a functional purpose. The performance gains of team-then-trim are also substantial with up to 14.18% improvements over CLLM especially when dataset deficiencies exist, e.g., data incompleteness.
>
> >(2) Inconsistent results: while train-then-trim mostly outperforms other approaches, it is not universally the case. Understanding when it helps and when it doesn’t and why is important
>
> Team-then-trim is designed to address dataset deficiencies, e.g., imbalance, incompleteness, noise, and sparsity, and the framework yields its strongest improvements in these regimes. For example, in the data imbalance setting (label ratio LR:MR:HR = 7:2:1) with only 10 original samples, team-then-trim improves the AUC of the HR group by 8.32%. In the data incompleteness setting (label ratio LR:MR:HR = 8:2:0) with the same number of original samples, the improvement on the HR group increases to 14.18%. We can see performance gains are largest when the original dataset lacks coverage or contains structural distortions. When the original dataset is already sufficiently informative, e.g., larger sample sizes with mild noise, the marginal benefit of additional synthetic data naturally diminishes. In these “easy” regions, classical generators or even the original data alone already provide enough signal for stable training of downstream models, so team-then-trim may match but not dramatically exceed baselines. Importantly, these cases do not indicate instability of the framework, but rather reflect that the value of QC-filtered LLM generations is related to the severity of underlying deficiencies. Our data quality control process is designed to balance fidelity with exploration, retaining realistic, high-quality samples while also encouraging coverage of underrepresented or unseen regions. Consequently, when considering an evaluation metric that targets only one dimension of quality, e.g., $\alpha$-precision on the COMPAS dataset, team-then-trim may not achieve the top score. COMPAS is a dataset with well-documented fairness issues [1], which means its real distribution is highly concentrated and exhibits limited feature variability. In this case, our $\alpha$-precision is 0.47% lower than CLLM because $\alpha$-precision emphasizes high-fidelity reconstruction of the densest regions of the real distribution, whereas our diversity-oriented trimming deliberately trades a small amount of local fidelity for broader coverage and stronger alignment with downstream objectives. This trade-off is beneficial for downstream learning, as reflected in consistently stronger $\beta$-recall, detection scores, and predictive performance, indicating that team-then-trim prioritizes task-aligned, structurally diverse synthetic samples rather than optimizing fidelity alone. The method design of team-then-trim explains why improvements are not uniform across all settings and highlights a practical strength: team-then-trim adaptively provides large gains when needed, while remaining competitive when the original dataset is already adequate.

---

> ### Author Response · Authors · 2025-11-30
> **Official Rebuttal to Reviewer fpt4 (2/3)**
>
> >(3) Missing computational cost: given the extra LLM calls via multi-agent, it is important to understand the performance cost vs performance gain trade-off
>
> We thank the reviewer for the feedback. We added the analysis of computational costs in Appendix D of the revised manuscript (highlighted in blue) which are also presented here for the convenience of discussion: “To contextualize the computational costs of our framework, we examine the token usage and runtime of each LLM component during data generation under the diabetes data-imbalance setting. Because the QC stage relies on lightweight procedures and small models relative to our data size, its computational overhead is negligible compared with the cost of LLM-based generation. The final generated dataset has a dimensionality of 10 rows $\times$ 9 columns. As shown in Table 8, team-then-trim incurs moderately higher token consumption and runtime than CLLM; however, the framework remains cost-efficient given its substantially stronger downstream utility. Specifically, Table 1 shows that team-then-trim improves AUC by 7.12\% in the LR group, 4.88\% in the MR group, and 11.19\% in the HR group compared with CLLM. These gains highlight that structured LLM teaming does not introduce prohibitive overhead and, in fact, achieves a highly favorable cost–benefit tradeoff in data-deficiency regimes.”
>
> | **Method**         | **Components**  | **Tokens (Input)** | **Tokens (Output)** | **Time (s)** |
> |--------------------|-----------------|---------------------|----------------------|--------------|
> | CLLM               | --              | 771                 | 872                  | 6.56         |
> | Team-then-Trim| LLM Worker 1    | 832                 | 387                  | 4.83         |
> |                    | LLM Worker 2    | 1831                | 887                  | 6.08         |
> |                    | LLM Worker 3    | 2457                | 1483                 | 7.16         |
> |                    | LLM Worker 4    | 2533                | 1507                 | 7.46         |
>
> >(4) Source of gain: It’s useful to understand if the source of gain is from the multi-agent generation or the QC approach. What if the CLLM curation mechanism were applied to the multi-agent generation, how would it perform with the new QC mechanism? i.e. an important ablation is Train-then-team + CLLM curation, CLLM generation + new QC mechanism
>
> Our results show that both LLM-teaming generation and the QC pipeline contribute distinct gains. First, LLM-teaming generation already provides improvements even without QC. Across Table 1-4, team-then-trim w/o QC mostly outperforms CLLM with curation. This indicates that the structured, dependency-aware generation from LLM teaming is itself a major source of gain. Second, Table 6 in the revised manuscript (also presented below) shows that each QC stage incrementally improves AUC under the data-imbalance setting of simulated Diabetes dataset. Together, the three stages refine raw generations into data that is simultaneously valid, task-aligned, and diverse, yielding the highest AUC across all groups
>
> | Methods                           | LR   | MR   | HR   |
> |-----------------------------------|------|------|------|
> | $D_{ori}$                         | 0.64 | 0.59 | 0.64 |
> | CLLM                              | 0.64 | 0.57 | 0.66 |
> | Team-then-Trim (w/o QC)           | 0.66 | 0.57 | 0.76 |
> | Team-then-Trim (after QC Step 1)  | 0.66 | 0.57 | 0.76 |
> | Team-then-Trim (after QC Step 2)  | 0.68 | 0.59 | 0.76 |
> | Team-then-Trim                    | 0.72 | 0.62 | 0.77 |

---

> ### Author Response · Authors · 2025-11-30
> **Official Rebuttal to Reviewer fpt4 (3/3)**
>
> >(5) Analysis of teaming — more in-depth analysis on when teaming provides value. What types of datasets, what dataset sizes, when should it be used and when does it provide minimal value
>
> Our study reveals that LLM teaming is most valuable when the dataset exhibits structural deficiencies, i.e., low sample size, missing or rare subpopulations, imbalanced distributions, or noisy labels, and provides comparatively smaller gains when the original data already offers broad and stable coverage of the underlying distribution. Also, as shown in Figure 10, LLM teaming provides value for datasets whose feature space exhibits nontrivial structure, e.g., when features interact through dependencies, partial causal pathways, or multi-level semantic groupings. In such datasets, generating all columns with a single LLM often produces inconsistencies or unrealistic combinations, because the model must jointly reason over heterogeneous concepts (e.g., demographics, behavior, outcomes) within one prompt. By contrast, LLM teaming decomposes the data space into semantically coherent components and preserves their dependency structure. Across our simulated experiments and real-world datasets, we observe that LLM teaming significantly improves downstream model utility when the number of real samples is very small or even when certain subpopulations contain no samples at all. For example, in the data incompleteness setting with only 10 original samples (8:2:0), teaming alone (without QC) boosts HR-group AUC by 11.85% compared with single LLM. In data imbalance setting (7:2:1), the AUC improvement on the HR-group is 10.55%. LLM teaming also yields robust gains under moderate and severe noise. Figures 3a-c and Table 3 show that team-then-trim maintains stable AUC improvements especially when flip ratio is 0.3 and 0.4. Structured generation is less prone to amplifying noise because LLM workers can generate coherent clusters of features.
>
> >(6) Additional LLMs: the paper only uses Llama as the backbone LLM. It is important to try different architectures of LLMs, different sizes and more recent LLMs, such that we know as of today if multi-agent is needed and for all LLMs.
>
> We added new experiments using Grok 4.1 Fast under both data-imbalance and data-incompleteness settings (see Tables 1 and 2 of the revised manscript, which are also presented here for the convenience of discussion). The results show that team-then-trim continues to deliver consistent improvements across backbone types.
>
> ### AUC under Data Imbalanced Setting
> | **Method** | **LLM Backbone** | **LR** | **MR** | **HR** |
> |-----------|------------------|--------|--------|--------|
> | CLLM | Grok 4.1 Fast | 64.29 | 54.72 | 66.87 |
> | Team-then-Trim (w/o QC) | Grok 4.1 Fast | 65.89 | 56.64 | 75.90 |
> | Team-then-Trim | Grok 4.1 Fast | 66.04 | 56.67 | 76.45 |
>
> ### AUC under Data Incompleteness Setting
> | **Method** | **LLM Backbone** | **LR** | **MR** | **HR** |
> |-----------|------------------|--------|--------|--------|
> | CLLM | Grok 4.1 Fast | 82.58 | 48.99 | 53.68 |
> | Team-then-Trim (w/o QC) | Grok 4.1 Fast | 82.44 | 51.70 | 56.50 |
> | Team-then-Trim | Grok 4.1 Fast | 82.98 | 52.59 | 58.13 |
>
> >Please can you had the computational costs and number of LLM calls
>
> Please see our response to Comment (3).
>
> >Please can you add analysis with other LLMs bases, sizes/parameters (and generally more recent)
>
> Please see our response to Comment (6).
>
> >Please can you add this abaltion to understand which component (generation vs curation) drives improvement? Team-then-Trim generation + CLLM's learning dynamics curation vs CLLM's single-LLM generation + Team-then-Trim's 3-stage QC
>
> Please see our response to Comment (4).
>
> >Please can you add std dev for all results over the 10 seeds
>
> We added the confidence intervals (CIs) in Table 6 of the manuscript. The CIs become slightly wider after QC because the trimming process reduces the number of retained samples and increases the variability across random seeds.
>
> >The datasets used are public and might be known to the LLM. Some analysis needs to be done on newer/private datasets to assess generalisation
>
> In addition to real-world public datasets, our paper evaluates team-then-trim extensively on two fully simulated datasets, each constructed from generative processes not present in public datasets: (1) a diabetes-risk dataset based on synthetic physiological and behavioral mechanisms, (2) a travel-behavior dataset generated from a latent decision threshold model with custom structural relationships. Because the ground-truth data-generating functions are privately constructed and mathematically defined, they serve as unseen datasets by design, allowing us to test generalization in a controlled setting.
>
> [1] Wang, Hanchen, et al. "An empirical study on learning fairness metrics for compas data with human supervision." arXiv preprint arXiv:1910.10255 (2019).

---

### Official Review · Reviewer_fEM3 · 2025-10-31

**Soundness:** 3
**Presentation:** 3
**Contribution:** 3
**Rating:** 4
**Confidence:** 4

**Summary:**

This paper proposes a framework “Team, then trim” which consists of aggregated workers (LLMs) specialized to (conditionally) generate specific (subset of) columns thus capturing the inter-feature logics and domain dependencies. Once the synthetic dataset has been generated, it undergoes a quality check (QC) process, namely sanity check, objective based cost assessment (by comparing synthetic samples with bootstrapped original data) and diversity based monitoring (to make sure the samples are not skewed) to make sure that the synthetic dataset is of high quality.

**Strengths:**

- Overall idea and analogy of assembly line workers is intuitive enough.
- The paper is clear to understand and well presented.

**Weaknesses:**

- [**Experiment on recent baselines**] Addition of more recent baselines, especially the ones that explored the usage of LLMs for tabular generation [1, 2, 3] will strengthen the paper. Moreover, ‘team-then-trim’ has some similarities with [1] in terms of using specialized model components per column/subset of columns (MoEs for [1], worker LLMs here), so it is also important to compare and contrast the pros and cons in related works.

- [**Experiment on model sizes**] Varying model sizes will be interesting to understand its importance on data quality. Questions around design choices such as “Larger/smaller task manager + smaller/larger role specialists” i.e do we need more capable task manager with average workers or an average task manager with highly capable workers; will be interesting to understand.
    - Questions like “how’s varying model size affects data quality” will also come under this experimental design choice.

- [**Experiment on model families**] Related to above and a follow-up one would be to look at different families of LLMs for Task Manager and role specialists. Will there be any bias coming up due to collaborations among LLMs coming from different families?
    - Eg: in LLM-as-a-judge literature, there’s bias associated with model preferring responses given by it’s family [4] i.e self-preference bias. So, any analysis and observation in that direction would be interesting.
    - Appendix E is an interesting starting point for addressing this kind of follow-up questions.

- [**Discussion on time complexity**] Please add time complexity analysis for proposed framework. The size of task manager and role specialist workers (if they are different), number of columns and rows to be generated, number of LLM workers to be used, costs associated with data quality checks (clustering) etc; contribute to overall time complexity. As some of it is data-specific (columns) and task-manager specific decisions (how many workers to assign), it is important to understand the time complexity from practical standpoint beforehand.
    - [Question, L804] Please add more details in this section in terms of how many samples were generated for each dataset?

- [**Discussion/Experiment on additional metrics**] Inclusion of additional metrics such as MLE (Machine Learning Efficacy) [5], DCR (Distance to closest record) [6], Discrimination [7] is important for discussions associated with privacy preservation, synthetic-vs-real data quality validation.
    - This will complement some of the discussions in Sec 2.2, especially for objective and diversity cost assessments.
    - Consider adding descriptions of various metrics in Appendix (including AUC, Accuracy, F1, Precision, Recall etc;) complementing sec 3.2.2.

- [**Discussion/Experiment on construction and evaluation of `G`**] From Fig 5 (L760-765), I see that task manager LLM is responsible in forming the relationships among data (i.e construction of `G`, eq. 1), and would like to know how it fairs with manual-human graph construction and assignment of worker LLMs. And how can one evaluate the quality of `G` i.e discard it or regenerate the work assignments.

- [**Discussion/Possible Experiment**] How can one extend the framework for their use case specific requirements for which LLMs doesn’t have enough domain knowledge, let’s say rare data which LLMs didn’t learn in their training process? For example, to generate UUIDs, distinct IDs which is rare/might be spurious from training process. Is it possible to do some fine-tuning with the current framework to get reliable predictions?

- [**Discussion/Experiment on column dependencies**] Following up from previous point, how does the conditional order of data generation affect in scenarios when columns has a bidirectional relationship i.e there can be different choices to resolve a scenario such as:
  - Generate column A, then column B vs
  - Generate column B, then column A or
  - Generate both A and B together. So, understanding how task-manager (LLM) and human might resolve role conflicts would be interesting.
    - A quick experiment would be to pick a dataset and have task-manager generated roles and human generated roles and compare the performance differences on role conflicts and worker assignment differences.
    - Consider adding discussion on different scenarios i.e “independent columns, unidirectionally causal columns and bidirectionally causal columns”.

1. Tabby: Tabular Data Synthesis with Language Models: https://arxiv.org/abs/2503.02152
2. Language Models are Realistic Tabular Data Generators: https://arxiv.org/abs/2210.06280
3. HARMONIC: Harnessing LLMs for Tabular Data Synthesis and Privacy Protection: https://arxiv.org/abs/2408.02927v1
4. Self-preference bias in LLM-as-a-judge: https://arxiv.org/abs/2404.13076, https://arxiv.org/abs/2410.21819
5. A Multi-Dimensional Evaluation of Synthetic Data Generators: https://ieeexplore.ieee.org/document/9686689
6. SynthEval: A Framework for Detailed Utility and Privacy Evaluation of Tabular Synthetic Data: https://arxiv.org/abs/2404.15821
7. Synthcity: facilitating innovative use cases of synthetic data in different data modalities: https://arxiv.org/abs/2301.07573

**Questions:**

- L462-463: Is there a hypothesis on why some metrics are not better for the proposed framework.
- L338: Is there a sentence continuing post “generated data”, I didn’t get that part.
- Consider adding a section describing the datasets used along with representative examples.

---

> ### Author Response · Authors · 2025-11-30
> **Official Rebuttal to Reviewer fEM3 (1/4)**
>
> Thank you for your valuable and insightful comments. Please see our response below.
>
> >[Experiment on recent baselines] Addition of more recent baselines, especially the ones that explored the usage of LLMs for tabular generation [1, 2, 3] will strengthen the paper. Moreover, ‘team-then-trim’ has some similarities with [1] in terms of using specialized model components per column/subset of columns (MoEs for [1], worker LLMs here), so it is also important to compare and contrast the pros and cons in related works.
>
> We thank the reviewer for the suggestion. We agree that these baselines represent meaningful advances in synthetic data generation. However, we did not include them as baselines because their operational assumptions differ fundamentally from those of team-then-trim, making a direct comparison unfair or misleading. Our framework is designed for scenarios where the original dataset is highly deficient, i.e., too small, imbalanced, incomplete, or noisy to support reliable fine-tuning of large models. Accordingly, we intentionally restrict ourselves to frozen, off-the-shelf pretrained LLMs, without any additional fine-tuning, domain adaptation, or training of large generative models, both to match the data-deficiency setting and to keep computational cost feasible. In contrast, the methods mentioned by the reviewer require substantially more real data, supervision, compute, or training overhead. For these reasons, we focus on baselines that also operate with frozen LLMs or non-parametric generators, ensuring a fair comparison under the same resource and data constraints.
>
> Tabby’s MoE assigns one expert per column, treating columns largely independently. In contrast, team-then-trim uses semantic grouping: the task manager partitions features into coherent components and sets dependency order. Workers condition on previously generated components to preserve inter-component logic, which is essential when domain constraints span multiple columns. This goes beyond Tabby’s column-wise independence. Also, because Tabby requires architectural modifications and LLM fine-tuning, it targets settings with moderate training data and compute budgets. Team-then-trim is designed for low-data, noisy and zero-fine-tuning settings, where Tabby’s training demands are not feasible and raw LLM generations remain inconsistent without QC.
>
> >[Experiment on model sizes] Varying model sizes will be interesting to understand its importance on data quality. Questions around design choices such as “Larger/smaller task manager + smaller/larger role specialists” i.e do we need more capable task manager with average workers or an average task manager with highly capable workers; will be interesting to understand. Questions like “how’s varying model size affects data quality” will also come under this experimental design choice.
>
> We appreciate the reviewer’s insightful comment. This is indeed an important and promising research direction. Fully exploring it, however, requires careful design along two dimensions: (1) Characterizing domain- or role-specific strengths of different LLMs, including how reliably they handle coordination, structural reasoning, or domain-specialized feature generation; and (2) Developing an assignment strategy that can efficiently map LLMs to roles under practical constraints, e.g., what to do when no LLM is strong in a domain, or when multiple LLMs show similar capabilities. A systematic evaluation along these two axes would require extensive modeling of LLM competencies, controlled role-swapping experiments, and potentially new metrics for role-specialization gains. We believe this constitutes a substantial research problem on its own and is therefore outside the scope of our current work. To ensure our conclusions do not depend narrowly on a single LLM family, we added new experiments using Grok 4.1 Fast under both data-imbalance and data-incompleteness settings (see Tables 1 and 2, which are also presented here for the convenience of discussion). The results show that team-then-trim continues to deliver consistent improvements across backbone types.
>
> ### AUC under Data Imbalanced Setting
> | **Method** | **LLM Backbone** | **LR** | **MR** | **HR** |
> |-----------|------------------|--------|--------|--------|
> | CLLM | Grok 4.1 Fast | 64.29 | 54.72 | 66.87 |
> | Team-then-Trim (w/o QC) | Grok 4.1 Fast | 65.89 | 56.64 | 75.90 |
> | Team-then-Trim | Grok 4.1 Fast | 66.04 | 56.67 | 76.45 |
>
> ### AUC under Data Incompleteness Setting
> | **Method** | **LLM Backbone** | **LR** | **MR** | **HR** |
> |-----------|------------------|--------|--------|--------|
> | CLLM | Grok 4.1 Fast | 82.58 | 48.99 | 53.68 |
> | Team-then-Trim (w/o QC) | Grok 4.1 Fast | 82.44 | 51.70 | 56.50 |
> | Team-then-Trim | Grok 4.1 Fast | 82.98 | 52.59 | 58.13 |

---

> ### Author Response · Authors · 2025-11-30
> **Official Rebuttal to Reviewer fEM3 (2/4)**
>
> >[Experiment on model families] Related to above and a follow-up one would be to look at different families of LLMs for Task Manager and role specialists. Will there be any bias coming up due to collaborations among LLMs coming from different families? Eg: in LLM-as-a-judge literature, there’s bias associated with model preferring responses given by it’s family [4] i.e self-preference bias. So, any analysis and observation in that direction would be interesting. Appendix E is an interesting starting point for addressing this kind of follow-up questions.
>
> Please see our response to [Experiment on model sizes].
>
> >[Discussion on time complexity] Please add time complexity analysis for proposed framework. The size of task manager and role specialist workers (if they are different), number of columns and rows to be generated, number of LLM workers to be used, costs associated with data quality checks (clustering) etc; contribute to overall time complexity. As some of it is data-specific (columns) and task-manager specific decisions (how many workers to assign), it is important to understand the time complexity from practical standpoint beforehand.
>
> We thank the reviewer for the feedback. We added the analysis of computational costs in Appendix D of the revised manuscript (highlighted in blue) which are also presented here for the convenience of discussion: “To contextualize the computational costs of our framework, we examine the token usage and runtime of each LLM component during data generation under the diabetes data-imbalance setting. Because the QC stage relies on lightweight procedures and small models relative to our data size, its computational overhead is negligible compared with the cost of LLM-based generation. The final generated dataset has a dimensionality of 10 rows $\times$ 9 columns. As shown in Table 8, team-then-trim incurs moderately higher token consumption and runtime than CLLM; however, the framework remains cost-efficient given its substantially stronger downstream utility. Specifically, Table 1 shows that team-then-trim improves AUC by 7.12\% in the LR group, 4.88\% in the MR group, and 11.19\% in the HR group compared with CLLM. These gains highlight that structured LLM teaming does not introduce prohibitive overhead and, in fact, achieves a highly favorable cost–benefit tradeoff in data-deficiency regimes.”
>
> | **Method**         | **Components**  | **Tokens (Input)** | **Tokens (Output)** | **Time (s)** |
> |--------------------|-----------------|---------------------|----------------------|--------------|
> | CLLM               | --              | 771                 | 872                  | 6.56         |
> | Team-then-Trim| LLM Worker 1    | 832                 | 387                  | 4.83         |
> |                    | LLM Worker 2    | 1831                | 887                  | 6.08         |
> |                    | LLM Worker 3    | 2457                | 1483                 | 7.16         |
> |                    | LLM Worker 4    | 2533                | 1507                 | 7.46         |
>
> >[Question, L804] Please add more details in this section in terms of how many samples were generated for each dataset?
>
> Across all experiments, the total number of generated samples is determined by the fixed batch size and the number of batches. **Diabetes**: batch size = 10. In imbalance and incompleteness experiments, there are 10 batches, i.e., 100 generated samples before QC. In varying data-size experiments, the number of synthetic data is from 10 to 100 before QC. **Drug**: batch size = 10. The number of batches before QC is set equal to the size of the original dataset; for example, with 10 original samples, one batch is generated. **TravelBehavior**: batch size = 20. In noise experiments, there are 20 batches before QC. In varying data-size experiments, the number of batches before QC is set equal to the size of the original dataset. **COMPAS**: batch size = 20. The number of batches before QC is set equal to the size of the original dataset.

---

> ### Author Response · Authors · 2025-11-30
> **Official Rebuttal to Reviewer fEM3 (3/4)**
>
> >[Discussion/Experiment on additional metrics] Inclusion of additional metrics such as MLE (Machine Learning Efficacy) [5], DCR (Distance to closest record) [6], Discrimination [7] is important for discussions associated with privacy preservation, synthetic-vs-real data quality validation. This will complement some of the discussions in Sec 2.2, especially for objective and diversity cost assessments.
>
> We thank the reviewer for the suggestion. In our work, the focus is on addressing data deficiencies (imbalance, incompleteness, noise, scarcity) rather than privacy preservation. Among the metrics you mentioned, Discrimination is already captured in our paper: the Detection score in Table 5 is a discriminator-based measure that evaluates how distinguishable synthetic samples are from real data, serving the same purpose as the Discrimination metric. MLE and DCR primarily assess memorization and record-level proximity, which are less aligned with our goal, since team-then-trim generates data via semantic LLM reasoning rather than instance-level perturbations.
>
> >Consider adding descriptions of various metrics in Appendix (including AUC, Accuracy, F1, Precision, Recall etc;) complementing sec 3.2.2.
>
> We added descriptions of our evaluation metrics in Appendix C, which are also copied here for convenience of discussion: “We use various evaluation metrics to capture different aspects of downstream predictive performance and quality of generated data. AUC measures the probability that a randomly chosen positive sample receives a higher predicted score than a randomly chosen negative sample. AUC is threshold-independent and particularly informative in imbalanced settings, which is why it serves as one of our primary metrics. Accuracy provides a holistic overview of performance but may be less informative under label imbalance. Recall measures the fraction of positive samples that are correctly identified. F1-score balances the trade-off between precision and recall and serves as a compact summary of classifier performance when neither error mode is dominant.”
>
> >[Discussion/Experiment on construction and evaluation of G] From Fig 5 (L760-765), I see that task manager LLM is responsible in forming the relationships among data (i.e construction of G, eq. 1), and would like to know how it fairs with manual-human graph construction and assignment of worker LLMs. And how can one evaluate the quality of G i.e discard it or regenerate the work assignments.
>
> We thank the reviewer for the feedback. Multiple valid graphs $G$ can be constructed for the same dataset, i.e., humans may produce different versions based on domain knowledge. Since our datasets do not contain ground-truth structural dependencies, quantitatively evaluating the correctness of $G$ is difficult. Qualitatively, the graphs produced by the task-manager LLM align well with human intuition. Most importantly, our QC pipeline reduces the impact of an imperfect $G$. If a graph leads to inconsistent or misaligned generations, these batches naturally fail the sanity check, objective-related cost assessment, or diversity inspection. Thus, the framework remains robust, and exploring automatic regeneration or selection of $G$ is a promising direction for future work.
>
> >[Discussion/Possible Experiment] How can one extend the framework for their use case specific requirements for which LLMs doesn’t have enough domain knowledge, let’s say rare data which LLMs didn’t learn in their training process? For example, to generate UUIDs, distinct IDs which is rare/might be spurious from training process. Is it possible to do some fine-tuning with the current framework to get reliable predictions?
>
> We thank the reviewer for the feedback. Our framework is modular, which means the LLM-teaming stage can be replaced or augmented with domain-specific generators whenever a feature requires exactness or rare-event support beyond the LLM’s pre-trained knowledge. Features where the LLM has enough pre-trained knowledge can continue to be generated by LLM workers. Features that require strict uniqueness, cryptographic randomness, or rare token patterns (e.g., UUIDs) can instead be generated by non-LLM modules, such as deterministic algorithms (e.g., UUIDv4 generators), domain-specific scripts, and rule-based or procedural samplers. We can also incorporate fine-tuning into team-then-trim, e.g., feature-specific fine-tuning, (i.e., only the LLM worker responsible for rare features needs fine-tuning, not the entire pipeline), LoRA [1] for low-data setting.

---

> ### Author Response · Authors · 2025-11-30
> **Official Rebuttal to Reviewer fEM3 (4/4)**
>
> >[Discussion/Experiment on column dependencies] Following up from previous point, how does the conditional order of data generation affect in scenarios when columns has a bidirectional relationship i.e there can be different choices to resolve a scenario such as: Generate column A, then column B vs Generate column B, then column A or Generate both A and B together. So, understanding how task-manager (LLM) and human might resolve role conflicts would be interesting. A quick experiment would be to pick a dataset and have task-manager generated roles and human generated roles and compare the performance differences on role conflicts and worker assignment differences. Consider adding discussion on different scenarios i.e “independent columns, unidirectionally causal columns and bidirectionally causal columns”.
>
> In team-then-trim, the task-manager LLM constructs dependency between components that naturally adapts to different relationship types, e.g., independent, unidirectional, and bidirectional, by generating independent columns in parallel, enforcing causal ordering when directionality is clear, and resolving bidirectional relationships by selecting a dominant direction grounded in semantic stability (e.g., demographic or structural attributes first) or combining these features into one component, while still conditioning later workers on all upstream outputs. This design ensures that even when the true relationship is symmetric or cyclic, the chosen ordering serves as a coherent approximation, and any residual misspecification is subsequently corrected by the QC pipeline. In particular, sanity checks remove value-level inconsistencies, objective-aligned trimming eliminates samples that break functional relationships relevant to prediction. Together, these QC stages act as a safeguard that filters out any batches introduced by imperfect ordering choices, ensuring that the final accepted batches preserve realistic joint patterns even in the presence of mutually dependent or ambiguous column relationships. This built-in robustness allows the framework to handle role-conflict scenarios without requiring manual intervention.
>
> >L462-463: Is there a hypothesis on why some metrics are not better for the proposed framework.
>
> In Table 5, $\alpha$-precision of team-then-trim on the COMPAS dataset is 0.47% lower than that of CLLM because $\alpha$-precision emphasizes high-fidelity reconstruction of the densest regions of the real distribution, whereas our framework intentionally trades a small amount of local fidelity for improved coverage and objective alignment. The QC pipeline is designed to admit batches that increase information gain and expand cluster-level diversity, which naturally encourages exploration beyond the tight, high-probability core modes favored by $\alpha$-precision. COMPAS is a dataset with well-documented fairness issues [2], which means its real distribution is highly concentrated and exhibits limited feature variability. Our diversity-oriented trimming can slightly pull samples away from the exact support of the majority mode, yielding marginally lower $\alpha$-precision. However, this trade-off is beneficial for downstream learning, as reflected in consistently stronger $\beta$-recall, detection scores, and predictive performance, indicating that team-then-trim prioritizes task-aligned, structurally diverse synthetic samples rather than optimizing fidelity alone.
>
> >L338: Is there a sentence continuing post “generated data”, I didn’t get that part.
>
> Thank you for pointing this out. What we intended to convey is that, for all reported results (except the baseline that uses $D_{ori}$ alone), the training data consists of the original dataset $D_{ori}$ combined with the generated data produced by our framework or by the baselines. We have revised the sentence to avoid ambiguity.
>
> >Consider adding a section describing the datasets used along with representative examples.
>
> In Figure5, the paragraph “Datasets simulation” and “Real-world datasets” of Appendix C in the revised manuscript, we presented more details and representative examples.
>
> [1] Hu, Edward J., et al. "Lora: Low-rank adaptation of large language models." ICLR 1.2 (2022): 3.
>
> [2] Wang, Hanchen, et al. "An empirical study on learning fairness metrics for compas data with human supervision." arXiv preprint arXiv:1910.10255 (2019).

---

### Official Review · Reviewer_cB7U · 2025-11-01

**Soundness:** 3
**Presentation:** 3
**Contribution:** 3
**Rating:** 4
**Confidence:** 3

**Summary:**

The paper proposes using an agentic AI approach for generating tabular data. A coordination LLM splits the generation problem into K parts assigned to different LLMs, each generating a subset of the tabular features in a way coordinated by the coordination LLM. Here, the prompt requires this LLM to handle dependencies between features for improved quality. The generated data is then passed into a three-stage quality check pipeline ensuring: 1) a sanity check for data types and values; 2) the provided learning potential for a given downstream model, and 3) a good level of diversity. The generated data is evaluated on the downstream task utility against baselines from related work.

**Strengths:**

- Leverages structural knowledge of the data during generation
- Incorporates multi-level quality checks to ensure high-quality data from different points software view: sanity, utility, and diversity
- Allows for the recovery of data subgroups missing in the original data

**Weaknesses:**

- Evaluation against related work misses typical tabular generators, e.g., GReaT [1] and Tabula [2], and in particular also any other agentic LLM, e.g., [3] or diffusion-based ones, e.g., [4].
- All LLMs in the evaluation seem to be of the same type, i.e., Llama 3.3 70B Instruct, but the power of this method could also be to use more targeted LLMs for the different roles, coordinator vs worker, or for specific features. No evaluation in this direction has been done.
- Following that, the same LLM is used for all roles, the paper should stress more what the advantage of this approach is with respect to some kind of chain-of-thought/in-context learning type of guidance of a single LLM during the data generation.
- Only full or no quality control is considered as an ablation study. It could be interesting how much each of the three QC steps contributes.

Minor:
- The type of data noise (label flip) has not been specified. Did you use symmetric or class-specific flipping?
- Table 1, 2, 4, 5: font is way larger than surrounding text.
- Figure 3 is placed before the text referencing it
- Figure 3: to ease comparison I suggest to use the same y range on all subfigures


[1] Borisov, V., Seßler, K., Leemann, T., Pawelczyk, M., Kasneci, G.: Language models are realistic tabular data generators. arXiv preprint arXiv:2210.06280 (2022)
[2] Zhao, Z., Birke, R., & Chen, L. Y. (2025, June). Tabula: Harnessing language models for tabular data synthesis. In Pacific-Asia Conference on Knowledge Discovery and Data Mining (pp. 247-259).
[3] Benoît Ronval, Pierre Dupont, Siegfried Nijssen. TAGAL: Tabular Data Generation using Agentic LLM Methods. arXiv preprint arXiv:/2509.04152 (2025)
[4] Akim Kotelnikov, Dmitry Baranchuk, Ivan Rubachev, Artem Babenko. TabDDPM: Modelling Tabular Data with Diffusion Models. ICML 2023: 17564-17579

**Questions:**

- How does Team-then-Trim perform against other baselines from related work, such as the ones referenced under weaknesses?
- What is the noise transition matrix used for label flipping?
- What is the benefit of the different QC steps?

---

> ### Author Response · Authors · 2025-11-30
> **Official Rebuttal to Reviewer cB7U (1/2)**
>
> Thank you for your valuable and insightful comments. Please see our response below.
> >(1) Evaluation against related work misses typical tabular generators, e.g., GReaT [1] and Tabula [2], and in particular also any other agentic LLM, e.g., [3] or diffusion-based ones, e.g., [4].
>
> Thank you for raising this important point. We agree that these baselines represent meaningful advances in synthetic data generation. However, we did not include them as baselines because their operational assumptions differ fundamentally from those of team-then-trim, making a direct comparison unfair or misleading. Our framework is designed for scenarios where the original dataset is highly deficient, i.e., too small, imbalanced, incomplete, or noisy to support reliable fine-tuning of large models. Accordingly, we intentionally restrict ourselves to frozen, off-the-shelf pretrained LLMs, without any additional fine-tuning, domain adaptation, or training of large generative models, both to match the data-deficiency setting and to keep computational cost feasible. In contrast, the methods mentioned by the reviewer require substantially more real data, supervision, compute, or training overhead. For these reasons, we focus on baselines that also operate with frozen LLMs or non-parametric generators, ensuring a fair comparison under the same resource and data constraints.
>
> >(2) All LLMs in the evaluation seem to be of the same type, i.e., Llama 3.3 70B Instruct, but the power of this method could also be to use more targeted LLMs for the different roles, coordinator vs worker, or for specific features. No evaluation in this direction has been done.
>
> We appreciate the reviewer’s insightful comment. This is indeed an important and promising research direction. Fully exploring it, however, requires careful design along two dimensions: (1) Characterizing domain- or role-specific strengths of different LLMs, including how reliably they handle coordination, structural reasoning, or domain-specialized feature generation; and (2) Developing an assignment strategy that can efficiently map LLMs to roles under practical constraints, e.g., what to do when no LLM is strong in a domain, or when multiple LLMs show similar capabilities. A systematic evaluation along these two axes would require extensive modeling of LLM competencies, controlled role-swapping experiments, and potentially new metrics for role-specialization gains. We believe this constitutes a substantial research problem on its own and is therefore outside the scope of our current work. To ensure our conclusions do not depend narrowly on a single LLM family, we added new experiments using Grok 4.1 Fast under both data-imbalance and data-incompleteness settings (see Tables 1 and 2, which are also presented here for the convenience of discussion). The results show that team-then-trim continues to deliver consistent improvements across backbone types.
>
> ### AUC under Data Imbalanced Setting
> | **Method** | **LLM Backbone** | **LR** | **MR** | **HR** |
> |-----------|------------------|--------|--------|--------|
> | CLLM | Grok 4.1 Fast | 64.29 | 54.72 | 66.87 |
> | Team-then-Trim (w/o QC) | Grok 4.1 Fast | 65.89 | 56.64 | 75.90 |
> | Team-then-Trim | Grok 4.1 Fast | 66.04 | 56.67 | 76.45 |
> ### AUC under Data Incompleteness Setting
> | **Method** | **LLM Backbone** | **LR** | **MR** | **HR** |
> |-----------|------------------|--------|--------|--------|
> | CLLM | Grok 4.1 Fast | 82.58 | 48.99 | 53.68 |
> | Team-then-Trim (w/o QC) | Grok 4.1 Fast | 82.44 | 51.70 | 56.50 |
> | Team-then-Trim | Grok 4.1 Fast | 82.98 | 52.59 | 58.13 |
>
>
>
> >(3) Following that, the same LLM is used for all roles, the paper should stress more what the advantage of this approach is with respect to some kind of chain-of-thought/in-context learning type of guidance of a single LLM during the data generation.
>
> Our LLM teaming decomposes the task into semantically coherent subtasks, which reduces the prompt complexity and allows the LLM to generate more consistent and domain-aligned data. Although we use the same LLM backbone, the role-specific prompting enforces separation of tasks, e.g., the label generator only reasons about outcome distribution, while other workers focus strictly on semantically related subsets of predictors. This is difficult to achieve with a single LLM with CoT or ICL.

---

> ### Author Response · Authors · 2025-11-30
> **Official Rebuttal to Reviewer cB7U (2/2)**
>
> >(4) Only full or no quality control is considered as an ablation study. It could be interesting how much each of the three QC steps contributes.
>
> We thank the reviewer for the suggestion. We added the AUC after each QC step in Table 6 of the revised manuscript, which is also presented here for the convenience of discussion: “Table 6 shows that each QC stage incrementally improves AUC under the data-imbalance setting of simulated Diabetes dataset. The LLM teaming output (w/o QC) already enhances HR performance by expanding coverage of rare cases, but offers limited gains for LR and MR due to noisy or misaligned samples. Sanity check (Step 1) yields no change, consistent with the fact that LLM teaming rarely violates basic constraints. Objective-related cost assessment (Step 2) produces the first clear improvement by removing samples with poor predictive alignment, which increases AUC in both LR and MR. Diversity-related monitoring (Step 3) delivers the final boost by admitting only batches that enhance cluster-level coverage. Together, the three stages refine raw generations into data that is simultaneously valid, task-aligned, and diverse, yielding the highest AUC across all groups.”
>
> | Methods                           | LR   | MR   | HR   |
> |-----------------------------------|------|------|------|
> | $D_{ori}$                         | 0.64 | 0.59 | 0.64 |
> | CLLM                              | 0.64 | 0.57 | 0.66 |
> | Team-then-Trim (w/o QC)           | 0.66 | 0.57 | 0.76 |
> | Team-then-Trim (after QC Step 1)  | 0.66 | 0.57 | 0.76 |
> | Team-then-Trim (after QC Step 2)  | 0.68 | 0.59 | 0.76 |
> | Team-then-Trim                    | 0.72 | 0.62 | 0.77 |
>
> >(5) The type of data noise (label flip) has not been specified. Did you use symmetric or class-specific flipping?
>
> Thank you for pointing this out. We used symmetric label flipping in all noise experiments. Specifically, for a given flip ratio, we uniformly sample a subset of data points without conditioning on class and randomly flip their labels. Thus, every class is equally likely to be corrupted, and no class receives preferential or targeted noise.
>
> >(6) Table 1, 2, 4, 5: font is way larger than surrounding text. Figure 3 is placed before the text referencing it. Figure 3: to ease comparison I suggest to use the same y range on all subfigures
>
> We have revised them.
>
> >(7) How does Team-then-Trim perform against other baselines from related work, such as the ones referenced under weaknesses?
>
> Please see our response to Comment (1).
>
> >(8) What is the noise transition matrix used for label flipping?
>
> Please see our response to Comment (5)
>
> >(9) What is the benefit of the different QC steps?
>
> Please see our response to Comment (4)

---

### Official Review · Reviewer_uPP4 · 2025-11-01

**Soundness:** 3
**Presentation:** 3
**Contribution:** 2
**Rating:** 4
**Confidence:** 3

**Summary:**

This paper introduces Team-then-Trim, a framework for synthetic tabular data generation using coordinated large language models. A task-manager LLM partitions the feature space into semantically aligned components and schedules specialized worker LLMs to generate each subset sequentially based on dependency structure. The resulting partial outputs are concatenated into full samples, which then pass through a three-stage quality control process assessing validity, task utility, and diversity preservation. Across simulated and real-world datasets, the proposed method yields synthetic data that improves downstream model performance and maintains distributional fidelity compared to both traditional oversampling and single-LLM baseline.

**Strengths:**

- The team-then-trim structure separates generation from post-hoc quality control, providing robustness against LLM hallucination.
- The three-stage quality control pipeline (sanity, objective-driven filtering, diversity enforcement) is systematic and targets well-known challenges in synthetic data generation, including invalid entries, distributional bias, and limited incremental information.
- The use of model-based scoring and information-gain comparison to filter batches offers a principled framework beyond heuristic rejection rules that previous work used.
- The method demonstrates downstream performance better than existing tabular data generation baselines.

**Weaknesses:**

- The quality control pipeline assumes access to a reasonably performant base model and sufficient initial real data to bootstrap quality signals, which can limit applicability in low-data or scarce-label settings (including simulated data incompleteness setting in the paper).
- The method incurs non-trivial computational overhead due to repeated generation, batch scoring, and rejection loops. The generation resource trade-offs are not fully addressed.
- The reliance on a single trained classifier for qualifying the cost of synthetic data raises the possibility that the QC process overfits to the specific classifier used, rather than reflecting true data utility. It would be valuable to evaluate whether the selected batches remain consistent when multiple different classifiers are used for the scoring stage.

**Questions:**

- The evaluation reports performance using 500 generated and original samples. How does downstream performance scale as the number of synthetic samples increases? Specifically, does performance continue to improve with additional synthetic data, or does it plateau or degrade?
- In scenarios where the number of original samples is limited, can the synthetic data still recover or cover the full cluster structure that would be observed if the complete real dataset were available? In other words, does the proposed method retain the ability to approximate the true distributional clusters when starting from a partially observed dataset?
- Which LLM was used for the curated generation process in CLLM? The original CuratedLLM paper reports that stronger LLMs exhibit better performance, particularly on under-represented samples. Therefore, it would be helpful to clarify the specific model used in your reproduction to understand the reported results.
- The proposed pipeline appears to rely on data-specific prompt construction for effective synthetic sample generation. Could the authors evaluate the robustness of the method with respect to prompt variations? Such an analysis would strengthen the novelty claim by demonstrating that performance is not overly reliant on manually curated prompt engineering.

---

> ### Author Response · Authors · 2025-11-30
> **Official Rebuttal to Reviewer uPP4 (1/2)**
>
> Thank you for your valuable and insightful comments. Please see our response below.
> >The quality control pipeline assumes access to a reasonably performant base model and sufficient initial real data to bootstrap quality signals, which can limit applicability in low-data or scarce-label settings (including simulated data incompleteness setting in the paper).
>
> Team-then-trim is explicitly designed for severe low-data and scarce-label conditions. Our experiments already use regimes with only 10 samples including cases where entire high-risk (HR) groups are missing, and team-then-trim improves the AUC on the HR group by 14.18% compared with baseline CLLM. The QC pipeline retains only those generated batches that demonstrably improve the quality of the original data, where “quality” is always evaluated in comparison to the original set via consistency checks, information-gain signals, and diversity improvements. Thus, team-then-trim remains applicable even when the base model is weak and the initial data are extremely limited.
>
> > The method incurs non-trivial computational overhead due to repeated generation, batch scoring, and rejection loops. The generation resource trade-offs are not fully addressed.
>
> We thank the reviewer for the feedback. We added the analysis of computational costs in Appendix D of the revised manuscript (highlighted in blue) which are also presented here for the convenience of discussion: “To contextualize the computational costs of our framework, we examine the token usage and runtime of each LLM component during data generation under the diabetes data-imbalance setting. Because the QC stage relies on lightweight procedures and small models relative to our data size, its computational overhead is negligible compared with the cost of LLM-based generation. The final generated dataset has a dimensionality of 10 rows $\times$ 9 columns. As shown in Table 8, team-then-trim incurs moderately higher token consumption and runtime than CLLM; however, the framework remains cost-efficient given its substantially stronger downstream utility. Specifically, Table 1 shows that team-then-trim improves AUC by 7.12\% in the LR group, 4.88\% in the MR group, and 11.19\% in the HR group compared with CLLM. These gains highlight that structured LLM teaming does not introduce prohibitive overhead and, in fact, achieves a highly favorable cost–benefit tradeoff in data-deficiency regimes.”
> | **Method**         | **Components**  | **Tokens (Input)** | **Tokens (Output)** | **Time (s)** |
> |--------------------|-----------------|---------------------|----------------------|--------------|
> | CLLM               | --              | 771                 | 872                  | 6.56         |
> | Team-then-Trim| LLM Worker 1    | 832                 | 387                  | 4.83         |
> |                    | LLM Worker 2    | 1831                | 887                  | 6.08         |
> |                    | LLM Worker 3    | 2457                | 1483                 | 7.16         |
> |                    | LLM Worker 4    | 2533                | 1507                 | 7.46         |
>
> >The reliance on a single trained classifier for qualifying the cost of synthetic data raises the possibility that the QC process overfits to the specific classifier used, rather than reflecting true data utility. It would be valuable to evaluate whether the selected batches remain consistent when multiple different classifiers are used for the scoring stage.
>
> In our framework, the objective-related cost assessment indeed assumes access to a downstream model. Our cost assessment selects only the middle quantile of samples (Eq. 6) rather than aggressively minimizing model residuals. This quantile-based selection prevents overfitting to any classifier used during QC, and in practice encourages selecting stable, low-variance samples aligned with the true signal rather than classifier-specific samples. As stated in the paper, we run the QC process and compute downstream performance separately for four heterogeneous models, i.e., Logistic Regression, SVM, MLP, and Random Forest. This means that for each model family, QC is conducted using that model’s predictive signal, and the final reported performance reflects the average results across all four learners. Because these models span linear, kernel-based, neural, and ensemble families, the fact that admitted batches consistently improve performance across all of them strongly suggests that QC is not overfitting to any kind of single classifier. While extending QC to use multiple models simultaneously is a meaningful direction, our existing results already demonstrate consistency and robustness across different classifier types.

---

> ### Author Response · Authors · 2025-11-30
> **Official Rebuttal to Reviewer uPP4 (2/2)**
>
> >The evaluation reports performance using 500 generated and original samples. How does downstream performance scale as the number of synthetic samples increases? Specifically, does performance continue to improve with additional synthetic data, or does it plateau or degrade?
>
> We would like to clarify that 500 refers to the number of test samples. The size of the generated dataset is not fixed and varies across experiments (see Appendix C: Data generation). In particular, as the size of the original dataset increases, the amount of generated data before QC also increases proportionally. Regarding how downstream performance scales with more synthetic data: performance does not follow a single monotonic trend, because the impact of additional synthetic samples depends on the dataset deficiency. As illustrated in Figure 3 and Figure 4, performance typically improves in the early stages but plateaus once the augmented data sufficiently saturates the decision boundary in low-noise settings. In higher-noise regimes, the performance plateaus at larger synthetic data sizes. Overall, these trends indicate that the utility of additional synthetic data is context-dependent rather than uniformly increasing.
>
> >In scenarios where the number of original samples is limited, can the synthetic data still recover or cover the full cluster structure that would be observed if the complete real dataset were available? In other words, does the proposed method retain the ability to approximate the true distributional clusters when starting from a partially observed dataset?
>
> Our framework is designed precisely to recover missing or underrepresented regions of the data space even when the original dataset provides only a partial view of the true distributional clusters. As shown in Figure 9, the generated samples populate the underlying cluster structure more fully than the original data alone, i.e., without HR group. In Table 1 and Table 2, in the extreme low-data settings where the original dataset contains only 10 samples with ratios LR:MR:HR = 7:2:1 or 8:2:0, the test set preserves the true cluster distribution, yet our method still achieves substantial AUC gains in both the MR and HR groups compared with original data. These improvements indicate that team-then-trim is able to reconstruct and cover the missing or sparsely observed clusters, despite starting from a highly limited and biased subset of the real data.
>
> > Which LLM was used for the curated generation process in CLLM? The original CuratedLLM paper reports that stronger LLMs exhibit better performance, particularly on under-represented samples. Therefore, it would be helpful to clarify the specific model used in your reproduction to understand the reported results.
>
> We use Llama 3.3 70B Instruct, the same backbone used in the team-then-trim for a fair comparison. In Tables 1 and 2, we added new experiments using Grok 4.1 Fast under both data-imbalance and data-incompleteness settings. Because the parameters and architecture of Grok 4.1 Fast are not publicly released, a direct comparison to Llama 3.3 70B Instruct is not available. However, the results consistently show that team-then-trim achieves improvements across both backbone families, demonstrating that our gains stem from the structured generation and QC pipeline rather than dependence on a specific LLM.
>
> > The proposed pipeline appears to rely on data-specific prompt construction for effective synthetic sample generation. Could the authors evaluate the robustness of the method with respect to prompt variations? Such an analysis would strengthen the novelty claim by demonstrating that performance is not overly reliant on manually curated prompt engineering.
>
> We thank the reviewer for the suggestion. As illustrated in Figures 6-8, the prompts in our framework follow a fixed and modular template that specifies task instructions, feature semantics, the original data excerpt, and output requirements, rather than relying on dataset-specific heuristics or hand-crafted prompt tuning. The task manager automatically constructs the dependency structure of components from the feature dictionary, and for different datasets or LLM workers, the varying elements in the prompt are (i) the role name assigned by the task manager, (ii) the feature names determined by the dataset, and (iii) the feature descriptions provided in the data dictionary. These pieces are inserted into the same prompt template in a systematic way, so no manual prompt redesign is needed when switching datasets. Moreover, even if minor prompt variations introduce stochastic deviations in the data generations, our three-stage QC pipeline rigorously evaluates each batch and filters out samples whose validity, objective alignment, or diversity deteriorates due to prompting noise. As a result, the framework remains stable and effective without relying on dataset-specific prompt engineering.

---

### Author Response · Authors · 2025-11-30
**General Response**

We would like to thank the reviewers for their valuable and insightful comments. Please see our point-to-point responses presented under each of the reviewers’ comments. We also significantly revised the manuscript according to the comments (please see the highlighted text in the revised manuscript). Below we summarize our major revisions and clarifications:

1. **Additional backbone LLMs.** We include new experiments using Grok 4.1 Fast under both data-imbalance and data-incompleteness settings. These results (Tables 1–2) show that team-then-trim consistently improves performance across backbone families, supporting the generality of the framework.

2. **Analysis of QC steps.** Responding to requests for a more granular ablation, we added Table 6 in Appendix D showing AUC after each QC component. This demonstrates that: (i) QC Step 1 (sanity check) shows stable validity. (ii) QC Step 2 (objective-related assessment) has first significant AUC gain. (iii) QC Step 3 (Diversity monitoring) achieves final improvement for all groups.

3. **Computational Cost.** We now provide a detailed token usage + runtime analysis in Table 8 of Appendix D. LLM teaming incurs moderate extra cost but yields substantial AUC gains (e.g., +14.18% in HR for data-incompleteness). QC overhead is negligible because QC relies on lightweight components. The resulting cost-benefit ratio is highly favorable.

4. **Relationship to recent baselines and positioning.** Several reviewers requested baselines such as GReaT, and other LLM systems. We explain that these methods require significantly more real data, fine-tuning, or heavy training, which violates our goal of operating in extreme data-deficiency settings. Our focus targets a different challenge, i.e., we aim to build a framework that works with extremely small, imbalanced, or incomplete datasets, for which reason we intentionally restrict ourselves to frozen, off-the-shelf pretrained LLMs, without any additional fine-tuning, domain adaptation, or training of large generative models, both to match the data-deficiency setting and to keep computational cost feasible. For fairness of comparison, we compare only with baselines that also operate with frozen LLMs or non-parametric generators under identical constraints.

We thank all reviewers again for their constructive feedback. We believe these revisions significantly strengthen the paper’s technical contributions, empirical validation, and practical relevance.

---

### Meta-Review · Area_Chair_j6JX · 2026-01-02

**Summary:**

This paper introduced a framework to generate tabular data by using a collaborative team of LLMs.  It contains a task manager LLMs to assign different worker LLMs (with a clearly defined role) to generate data. To control the data quality, it also introduces a three-stage quality control process.  It evaluates on simulated and real-world datasets to show the effectiveness.


Overall, reviewers initially gave 4, 4, 4. They shared some similar concerns about baselines, computational cost, and evaluations. AC read the rebuttal. AC feels this paper should evaluate the baseline proposed by the reviewers and also evaluate on more targeted LLMs with different teaming strategies, although the authors did some during the rebuttal. But overall, it is still not solid for ICLR. AC tends to suggest polishing the paper based on the reviewers' comments.

**Reviewer Concerns:**

Baseline comparison:
Some reviewers points out the concerns of the recent baselines. AC think for those heavily training based methods, it is ok to not include them. But Authors should compare the methods which use LLMs for tabular generation to better show the contributions.
Computational overhead:
AC feels it is ok to have reasonable overhead. But authors should add the performance comparison regarding the different components. Currently, the table only contains the overhead in regard to the components but missing the corresponding performance.
More backbones:
It is great to include grok 4.1 Fast. But why only grok 4.1 fast instead of others. More backbone should be included.

**Reviewer Scores:**

I feel most of the concerns have been addressed. However, there are some parts such as baseline are not fully address.

---

### Decision · Program_Chairs · 2026-01-26

Reject